# Introduction to the SPARC Reanalysis Intercomparison Project (S-RIP) and overview of the reanalysis systems

Masatomo Fujiwara[1], Jonathon S. Wright[2], Gloria L. Manney[3,4], Lesley J. Gray[5,6], James Anstey[7], Thomas Birner[8], Sean Davis[9,10], Edwin P. Gerber[11], V. Lynn Harvey[12], Michaela I. Hegglin[13], Cameron R. Homeyer[14], John A. Knox[15], Kirstin Krüger[16], Alyn Lambert[17], Craig S. Long[18], Patrick Martineau[19], Andrea Molod[20], Beatriz M. Monge-Sanz[21], Michelle L. Santee[17], Susann Tegtmeier[22], Simon Chabrillat[23], David G. H. Tan[21], David R. Jackson[24], Saroja Polavarapu[25], Gilbert P. Compo[10,26], Rossana Dragani[21], Wesley Ebisuzaki[18], Yayoi Harada[27,28], Chiaki Kobayashi[28], Will McCarty[20], Kazutoshi Onogi[27], Steven Pawson[20], Adrian Simmons[21], Krzysztof Wargan[20,29], Jeffrey S. Whitaker[26], and Cheng-Zhi Zou[30]

[1]Faculty of Environmental Earth Science, Hokkaido University, Sapporo, 060-0810, Japan
[2]Center for Earth System Science, Tsinghua University, Beijing, 100084, China
[3]NorthWest Research Associates, Socorro, NM 87801, USA
[4]Department of Physics, New Mexico Institute of Mining and Technology, Socorro, NM 87801, USA
[5]Atmospheric, Oceanic and Planetary Physics, University of Oxford, Oxford, OX1 3PU, UK
[6]NERC National Centre for Atmospheric Science (NCAS), Leeds, LS2 9JT, UK
[7]Canadian Centre for Climate Modelling and Analysis, Environment and Climate Change Canada, University of Victoria, Victoria, V8W 2Y2, Canada
[8]Department of Atmospheric Science, Colorado State University, Fort Collins, CO 80523, USA
[9]Earth System Research Laboratory, National Oceanic and Atmospheric Administration, Boulder, CO 80305, USA
[10]Cooperative Institute for Research in Environmental Sciences, University of Colorado at Boulder, Boulder, CO 80309, USA
[11]Courant Institute of Mathematical Sciences, New York University, New York, NY 10012, USA
[12]Laboratory for Atmospheric and Space Physics, University of Colorado, Boulder, CO 80303, USA
[13]Department of Meteorology, University of Reading, Reading, RG6 6BB, UK
[14]School of Meteorology, University of Oklahoma, Norman, OK 73072, USA
[15]Department of Geography, University of Georgia, Athens, GA 30602, USA
[16]Department of Geosciences, University of Oslo, Oslo, 0315, Norway
[17]Jet Propulsion Laboratory, California Institute of Technology, Pasadena, CA 91109, USA
[18]Climate Prediction Center, National Centers for Environmental Prediction, National Oceanic and Atmospheric Administration, College Park, MD 20740, USA
[19]Department of Atmospheric and Oceanic Sciences, University of California Los Angeles, Los Angeles, California, CA 90095, USA
[20]Global Modeling and Assimilation Office, Code 610.1, NASA Goddard Space Flight Center, Greenbelt, MD 20771, USA
[21]European Centre for Medium-Range Weather Forecasts, Shinfield Park, Reading, RG2 9AX, UK
[22]GEOMAR Helmholtz Centre for Ocean Research Kiel, Kiel, 24105, Germany
[23]Royal Belgian Institute for Space Aeronomy (BIRA-IASB), Brussels, 1180, Belgium
[24]Met Office, FitzRoy Road, Exeter, EX1 3PB, UK
[25]Climate Research Division, Environment and Climate Change Canada, Toronto, Ontario, M3H 5T4, Canada
[26]Physical Sciences Division, Earth System Research Laboratory, National Oceanic and Atmospheric Administration, Boulder, CO 80305, USA
[27]Japan Meteorological Agency, Tokyo, 100-8122, Japan
[28]Climate Research Department, Meteorological Research Institute, JMA, Tsukuba, 305-0052, Japan
[29]Science Systems and Applications Inc., Lanham, MD 20706, USA

[30]Center for Satellite Applications and Research, NOAA/NESDIS, College Park, MD 20740, USA

*Correspondence to*: Jonathon S. Wright (jswright@tsinghua.edu.cn) and Masatomo Fujiwara (fuji@ees.hokudai.ac.jp)

**Abstract.** The climate research community uses atmospheric reanalysis data sets to understand a wide range of processes and variability in the atmosphere, yet different reanalyses may give very different results for the same diagnostics. The

Stratosphere–troposphere Processes And their Role in Climate (SPARC) Reanalysis Intercomparison Project (S-RIP) is a coordinated activity to compare reanalysis data sets using a variety of key diagnostics. The objectives of this project are to identify differences among reanalyses and understand their underlying causes, to provide guidance on appropriate usage of various reanalysis products in scientific studies, particularly those of relevance to SPARC, and to contribute to future improvements in the reanalysis products by establishing collaborative links between reanalysis centres and data users. The

project focuses predominantly on differences among reanalyses, although studies that include operational analyses and studies comparing reanalyses with observations are also included when appropriate. The emphasis is on diagnostics of the upper troposphere, stratosphere, and lower mesosphere. This paper summarizes the motivation and goals of the S-RIP activity, and extensively reviews key technical aspects of the reanalysis data sets that are the focus of this activity. The special issue of "The SPARC Reanalysis Intercomparison Project (S-RIP)" in this journal serves to collect research with

relevance to S-RIP in preparation for the publication of the planned two (interim and full) S-RIP reports.

## 1 Introduction

An atmospheric reanalysis system consists of a global forecast model, input observations, and an assimilation scheme, which are used in combination to produce best estimates (analyses) of past atmospheric states (including temperature, wind, geopotential height, and humidity fields, among others). The forecast model propagates information forward in time and

space from previous analyses of the atmospheric state. The assimilation scheme then blends the resulting short-range forecast outputs with input observations to produce subsequent analyses of the atmospheric state, which are in turn used to initialize further forecasts. Whereas operational analysis systems are continuously updated with the intention of improving numerical weather predictions, reanalysis systems are fixed throughout their lifetime. Using a fixed assimilation–forecast model system to produce analyses of observational data previously analysed in the context of operational forecasting (the

"re" in "reanalysis") helps to prevent the introduction of artificial changes in the analysed fields (Trenberth and Olson, 1988; Bengtsson and Shukla, 1988), although artificial changes still arise from other sources (especially from changes in the quality and/or quantity of the input observational data). The first three major reanalysis efforts started in the late 1980s, conducted by NASA, ECMWF, and a joint effort between the NMC (now NCEP) and NCAR (e.g. Edwards, 2010). More than 10 global atmospheric reanalysis data sets are currently available worldwide. A key for all acronyms used in this paper

is provided in Appendix A. Acronyms representing the names of institutes, models, satellites, and other entities are in most cases only provided in the appendix; all other acronyms are both introduced in the text and included in the appendix.

Stratosphere-troposphere Processes And their Role in Climate (SPARC) is one of four core projects of the WCRP, and is sponsored by the WMO, ICSU, and IOC of UNESCO. Research themes within the SPARC mandate include atmospheric dynamics and predictability, chemistry and climate, and long-term records for understanding climate. Reanalysis data sets feature prominently among the tools used by the SPARC community to understand atmospheric processes and variability, to validate chemistry-climate models, and to investigate and identify climate change (e.g. SPARC, 2002; Randel et al., 2004; SPARC, 2010; and references therein). However, there are known challenges for middle atmosphere analysis and reanalysis, including (but not limited to) smaller volumes of observational data available for assimilation, increases in noise and/or biases in the available observations with height, and unique aspects of middle atmospheric dynamics that influence the behaviour of background error covariances and other facets of the data assimilation system (e.g. Swinbank and O'Neill, 1994; Swinbank and Ortland 2003; Polavarapu et al., 2005; Rood, 2005; Polavarapu and Pulido, 2016). It has been more than 10 years since the last comprehensive intercomparison of reanalyses and related data sets in the middle atmosphere, the SPARC Intercomparison of Middle Atmosphere Climatologies (SPARC, 2002; Randel et al., 2004), and several new reanalyses have been released in the intervening years. That intercomparison and multiple subsequent studies have shown that different results may be obtained for the same diagnostic due to different technical details of the reanalysis systems, even amongst more recent reanalyses (a list of examples has been provided by Fujiwara et al., 2012; see also the contents of this special issue). The pervasive nature of these discrepancies creates a need for a new coordinated intercomparison of reanalysis data sets with respect to key diagnostics that can help to clarify the causes of these differences. The results of this intercomparison are intended to provide guidance on the appropriate usage of reanalysis products in scientific studies, particularly those of relevance to SPARC. The reanalysis community also benefits from coordinated user feedback, which helps to drive improvements in the next generation of reanalysis products. The SPARC Reanalysis Intercomparison Project (S-RIP) was initiated in 2011 to conduct a coordinated intercomparison of all major global atmospheric reanalyses (Fujiwara et al., 2012; Fujiwara and Jackson, 2013; Errera et al., 2015; see also http://s-rip.ees.hokudai.ac.jp/). The goals of S-RIP are: (1) to better understand the differences among current reanalysis products and their underlying causes; (2) to provide guidance to reanalysis data users by documenting the results of this reanalysis intercomparison; and (3) to create a communication platform between the SPARC community and the reanalysis centres that helps to facilitate future reanalysis improvements. Documentation will include both peer-reviewed papers and two S-RIP reports published as part of the SPARC report series, a full report scheduled for publication in 2018 and an electronic-only interim report to be published beforehand.

Figure 1 shows a schematic illustration of the atmosphere highlighting the processes and themes covered by S-RIP. The planned S-RIP reports consist of two parts. Chapters 1–4 introduce the project, describe the reanalysis systems, and provide intercomparisons of basic variables (temperature, winds, ozone, and water vapour). These chapters will constitute the entirety of the interim report and will also be updated and included in the subsequent full report. Chapters 5–12 will only be included in the full report. The chapters to be included only in the full report will be arranged according to, and focus on, different regions or processes within the atmosphere, including the Brewer-Dobson circulation, stratosphere–troposphere

dynamical coupling, upper tropospheric/lower stratospheric processes in the extratropics and tropics, the quasi-biennial oscillation (QBO) and tropical variability, lower stratospheric polar chemical processing and ozone loss, and dynamics and transport in the upper stratosphere and lower mesosphere (Fig. 1). Some important topics, such as gravity waves and transport processes, are sufficiently pervasive that related aspects will be distributed amongst several chapters.

The S-RIP project focuses predominantly on reanalyses, although some chapters of the planned reports will include diagnostics from operational analyses when appropriate. In addition to intercomparison of diagnostics calculated directly from reanalysis products, some chapters will include discussion of chemical transport model (CTM) and trajectory model simulations driven by different reanalysis data sets. Table 1 lists reanalysis data sets that are currently available and will be included in one or more chapters of the planned S-RIP full report. Many of the chapters focus primarily on newer reanalysis

systems that assimilate upper-air measurements and produce data at relatively high resolution (e.g. ERA-Interim, JRA-55, MERRA, MERRA-2, and CFSR/CFSv2). We also intend to include forthcoming reanalyses (e.g. ERA5) when they become available, and long-term reanalyses that assimilate only surface meteorological observations (e.g. NOAA-CIRES 20CR and ERA-20C) where appropriate. Some chapters of the planned reports will include comparisons with older reanalyses (NCEP-NCAR R1, NCEP-DOE R2, ERA-40, and JRA-25/JCDAS), because these products have been heavily used in the past and

are still being used for some studies, and because such comparisons can provide insight into the potential shortcomings of past research results. Other chapters will only include a subset of these reanalysis data sets, since some reanalyses have already been shown to perform poorly for certain diagnostics (e.g. Pawson and Fiorino, 1998; Randel et al., 2000; Manney et al., 2003, 2005; Birner et al., 2006; Monge-Sanz et al., 2007; Parrondo et al., 2007; Sakazaki et al., 2012; Lu et al., 2015; Martineau et al., 2016) or do not extend high enough in the atmosphere. The intercomparison period common to all chapters

of the planned S-RIP reports is 1980–2010. This period starts with the advent of high-frequency remotely sensed data (the "satellite era") and ends with the transition between CFSR and CFSv2 (see below). Some chapters will also consider the pre-satellite era before 1979 and/or include results for more recent years. Given the wide use of ERA-40 (which only extends to August 2002), separate intercomparisons for 1979—2002 are also considered for selected diagnostics.

     The special issue of "The SPARC Reanalysis Intercomparison Project (S-RIP)" in this journal serves to collect research

with relevance to S-RIP in preparation for the publication of the planned two (i.e., interim and full) S-RIP reports. The remainder of this paper contains overview material intended to reduce duplication in subsequent manuscripts in this special issue, and is organized as follows. Section 2 is a brief introduction to the 11 global atmospheric reanalyses listed in Table 1. Section 3 is an overview of key differences among reanalysis forecast models, with a particular focus on major physical parametrizations and boundary conditions. Section 4 is a basic description of data assimilation as implemented in current

reanalysis systems. Section 5 is a summary comparison of frequently assimilated input observations, focusing on five of the most recent reanalysis systems. Section 6 concludes the manuscript with an outline of the intended future evolution of the S-RIP activity.

## 2 Current reanalysis systems

In this paper, we divide reanalysis systems into three classes according to their observational inputs. "Full input" reanalyses are systems that assimilate surface and upper-air conventional and satellite data. "Conventional input" reanalyses are systems that assimilate surface and upper air conventional data, but do not assimilate satellite data. "Surface input" reanalyses are systems that assimilate surface data only, with upper air observations excluded. Some of the reanalysis centres also provide companion "AMIP-type" simulations, which do not assimilate any observational data and are constrained by applying a sea surface temperature analysis as a lower boundary condition on the atmospheric model. The following discussion also includes the term "satellite era", which refers to the period following 1979 (the first full year of TOVS availability) for which satellite data are relatively abundant, and the companion term "extended reanalysis", which refers to any reanalysis that provides data for the period before January 1979.

We note that the production of reanalyses often must be completed under strict deadlines. To meet these deadlines, most reanalyses have been executed in two or more distinct "streams", which are then combined (Fig. 2). Discontinuities in the time series of some analysed variables may occur when streams are joined. The potential impacts of these discontinuities should be considered (along with changes in assimilated observations described in Sections 5 and 6) when reanalysis variables are used for assessments of climate variability and/or trends.

### 2.1 ECMWF reanalyses

ERA-40 (Uppala et al., 2005) is an extended full input reanalysis covering 45 years from September 1957 through August 2002. ERA-40 was released by ECMWF in 2003 and represents an important improvement relative to the first generation of modern reanalysis systems, including FGGE (Bengtsson et al., 1982) and ERA-15 (Gibson et al., 1997). ERA-40 did not assimilate satellite data prior to January 1973. The ERA-40 reanalysis from September 1957 through December 1972 is therefore a conventional input reanalysis. ERA-40 products continue to be used in many studies that require long-term atmospheric data.

ERA-Interim (Dee et al., 2011), initially released by ECMWF in 2008, is a full input reanalysis of the satellite era that includes several corrections and modifications to the system used for ERA-40. In particular, ERA-Interim uses a 4D-Var data assimilation system, which makes more complete use of observations collected between analysis times than the 3D-FGAT (first guess at appropriate time) approach used in ERA-40 (see Section 4). Major focus areas during the production of ERA-Interim included achieving more realistic representations of the hydrologic cycle and the stratospheric circulation relative to ERA-40, as well as improving the consistency of the reanalysis products in time.

ERA-20C (Poli et al., 2015, 2016) is a surface input reanalysis produced by ECMWF and released in 2014. ERA-20C uses a 4D-Var data assimilation system, but takes its spatially and temporally varying background errors from a prior ensemble data assimilation (Isaksen et al., 2010; Poli et al., 2013). Because ERA-20C directly assimilates only surface pressure and surface wind observations, it can generate reanalyses of the climate state that extend further back in time (in this case to the beginning of the twentieth century). Assimilation of surface data indirectly constrains the upper-atmospheric

state, but these constraints are relatively weak on longer-than-synoptic timescales. While data from ERA-20C extend up to 0.01 hPa, these data should be used with caution in the upper troposphere and above. The ERA-20C model also uses sea surface temperature and sea ice concentration analyses, and radiative forcings prescribed for CMIP5. The companion product ERA-20CM (Hersbach et al., 2015) provides an ensemble of AMIP-style simulations using similar forcings and lower boundary conditions. Ensemble members are spun up from the same initial state, and differ only in the prescribed evolution of sea surface temperature (SST) and sea ice, which are drawn from the HadISST2 ensemble (Titchner and Rayner, 2014; Hersbach et al., 2015).

## 2.2 JMA reanalyses

JRA-25 (Onogi et al., 2007), released in 2006, is a full input reanalysis of the satellite era and the first reanalysis produced by JMA (in cooperation with CRIEPI). This reanalysis originally covered 25 years from 1979 through 2004, and was extended by an additional 10 years (through the end of January 2014) as JCDAS using an identical fixed model–assimilation system.

JRA-55 (Kobayashi et al., 2015), released in 2013, is an extended full input reanalysis produced by JMA. JRA-55 is the most recent reanalysis that both assimilates upper air observations and includes coverage of the pre-TOVS era (i.e. before November 1978), starting from the International Geophysical Year (IGY) in January 1958. To date, JRA-55 is the only reanalysis system to apply a 4D-Var data assimilation scheme to upper-air data during the pre-satellite era (ERA-20C has also applied 4D-Var, but only to surface observations). Two companion products are also available: JRA-55C (Kobayashi et al., 2014), a conventional input reanalysis, and JRA-55AMIP, an AMIP-style forecast model simulation without data assimilation. Both JRA-55C and JRA-55AMIP were released to the public in 2015. JRA-55C is available starting from November 1972, two months before JRA-55 began assimilating satellite observations (before this date, JRA-55 only assimilated conventional observations, so that JRA-55 and JRA-55C are identical), and extends through December 2012. JRA-55AMIP extends from January 1958 through December 2012. Extensions beyond December 2012 are planned for both JRA-55C and JRA-55AMIP, but details have not been determined as of this writing.

## 2.3 NASA GMAO reanalyses

MERRA (Rienecker et al., 2011), released in 2009, is a full input reanalysis of the satellite era developed by NASA's GMAO using the GEOS-5 data assimilation system. MERRA was conceived with the intention of leveraging the large amounts of data produced by NASA's Earth Observing System (EOS) satellite constellation and improving the representations of the water and energy cycles relative to earlier reanalyses. The top level used in MERRA (0.01 hPa, approximately 80 km) is higher than the top levels used in most other reanalyses, which facilitates studies extending into the mesosphere. An earlier NASA reanalysis (Schubert et al., 1993; Schubert et al., 1995) covering 1980–1995 was produced by NASA's DAO (now GMAO) using the GEOS-1 data assimilation system; this reanalysis is no longer publicly available, and is not included in the S-RIP intercomparison.

MERRA-2 (Bosilovich et al., 2015), released in 2015, is a full input reanalysis of the satellite era from NASA's GMAO. As the follow-on to MERRA, the production of MERRA-2 was motivated by the inability of the MERRA system to ingest some recent data types. MERRA-2 includes substantial upgrades to the model (Molod et al., 2015) and changes to the data assimilation system and input data. New constraints are applied to ensure conservation of global dry-air mass and to close the balance between surface water fluxes (precipitation minus evaporation) and changes in total atmospheric water (Takacs et al., 2016). Other new features in MERRA-2 relative to MERRA include a modified gravity wave scheme that substantially improves the model representation of the QBO (Molod et al., 2015; Coy et al., 2016); assimilation of MLS temperature retrievals at high altitudes (pressures less than or equal to 5 hPa) to better constrain the reanalysis at upper levels; assimilation of MLS stratospheric ozone profiles and OMI column ozone since the beginning of the Aura mission in late 2004 to improve representation of fine-scale ozone features, especially in the region around the tropopause; and assimilation of aerosol optical depth (AOD; Randles et al., 2016), with analysed aerosols fed back to the forecast model radiation scheme.

## 2.4 NOAA/NCEP and related reanalyses

NCEP–NCAR R1 (Kalnay et al., 1996; Kistler et al., 2001), which was first released in 1995, was the first modern reanalysis system with extended temporal coverage (1948–present), and was produced using a modified 1995 version of the NCEP forecast model. NCEP–DOE R2 (Kanamitsu et al., 2002), released in 2000, covers the satellite era (1979–present) using a 1998 version of the same model, and includes corrections for some important errors and limitations identified in R1. Both R1 and R2 remain in widespread use; however, these systems have relatively low top levels (3 hPa), relatively coarse vertical resolutions (28 levels), and assimilate retrieved temperatures rather than radiances from the operational nadir sounders, rendering them unsuitable for most studies of the middle atmosphere.

CFSR (Saha et al., 2010), released in 2009, is a full input reanalysis of the satellite era that uses a 2007 version of the NCEP CFS. CFSR contains a number of improvements relative to R1 and R2 in both the forecast model and data assimilation system, including higher horizontal and vertical resolutions, a higher model top, more sophisticated model physics, and the ability to assimilate satellite radiances directly. CFSR is also the first global reanalysis of the coupled atmosphere–ocean–sea ice system. Official data coverage by CFSR only extends through December 2009, but output from the same analysis system was continued through December 2010 before being migrated to the operational CFSv2 analysis system (Saha et al., 2014) from January 2011. This transition from CFSR to CFSv2 should not be confused with the transfer of CFSv2 production from NCEP EMC to NCEP operations, which occurred at the start of April 2011. CFSv2 has a different horizontal resolution and includes minor changes to physical parametrizations (some of which are described below), but is intended to serve as a continuation of CFSR and can be treated as such for most purposes. To distinguish CFSR and its CFSv2 continuation from other (mainly forecast) applications of the NCEP CFS that do not use the full data assimilation system, CFSR may also be referred to as CDAS-T382 and its continuation as CDAS-T574.

NOAA–CIRES 20CR v2 (Compo et al., 2011), released in 2009, is the first reanalysis to span more than 100 years. Like ERA-20C, 20CR is a surface input reanalysis. Unlike ERA-20C, which uses a 4D-Var approach to assimilate both surface pressure and surface winds, 20CR uses an ensemble Kalman filter (EnKF) approach (see Section 4) and assimilates only surface pressure. The forecast model used in 20CR is similar in many ways to that used in CFSR, but with much coarser vertical and horizontal resolutions. 20CR provides reanalysis fields back to the mid-nineteenth century. With only surface observations assimilated and a modest vertical resolution, 20CR is likely to be of limited utility for most studies above the tropopause and many in the upper troposphere.

## 3 Forecast model specifications

The forecast model is a fundamental component of any atmospheric reanalysis system. Major differences in forecast model specifications among current reanalysis systems include the horizontal grid type and spacing, the number of vertical levels, the height of the top level, the formulation of physical parametrizations, and the choice of various boundary conditions.

Table 2 provides the basic specifications for each of the reanalysis forecast models. Most of the models use spectral dynamical cores (e.g. Machenhauer, 1979), with the exception of MERRA and MERRA-2, which use finite-volume dynamics (Lin, 2004). The horizontal resolutions of the forecast models range from approximately 1.875° (R1, R2, and 20CR) to approximately 0.2° (CFSv2). A variety of notations have been used to describe Gaussian grids used in models based on spectral dynamical cores. Here, we use F$n$ to refer to the regular Gaussian grid with $2n$ latitude bands and (typically) $4n$ longitude bands. The longitude grid spacing in the standard F$n$ regular Gaussian grid is $90°/n$, so that the geographical distance between neighbouring grid cells in the east–west direction shrinks toward the poles. R1, R2, and 20CR use the same regular Gaussian grid (F47), which differs from the standard in that it has $4(n+1)$ longitude bands and a longitude spacing of $90°/(n+1)$. JRA-25 (F80), CFSR (F288), and CFSv2 (F440) also use regular Gaussian grids. ERA-Interim, ERA-40, ERA-20C, and the JRA-55 family use linear reduced Gaussian grids (Hortal and Simmons, 1991; Courtier and Naughton, 1994), which are denoted by N$n$. The number of latitude bands in the N$n$ reduced Gaussian grid is also $2n$, but the number of longitudes per latitude circle decreases from the equator (where it is $4n$) toward the poles. Longitude grid spacing in reduced Gaussian grids is therefore quasi-regular in distance rather than degrees. The effective horizontal grid spacing is approximately 79 km for ERA-Interim (N128), approximately 125 km for ERA-40 and ERA-20C (N80), and approximately 55 km for JRA-55 (N160). Latitude bands in both regular and reduced Gaussian grids are irregularly spaced and symmetric around the equator, with locations defined by the zeros of the Legendre polynomial of order $2n$. The horizontal resolution of a Gaussian grid may also be described via the wavenumber truncation. Wavenumber truncations for reanalysis forecast models using regular or reduced Gaussian grids are listed in Table 2. MERRA ($^1/_2°$ latitude $\times$ $^2/_3°$ longitude) and MERRA-2 ($^1/_2°$ latitude $\times$ $^5/_8°$ longitude) use regular latitude–longitude grids.

All of the reanalysis systems listed in Table 1 use hybrid $\sigma$–$p$ vertical coordinates (Simmons and Burridge, 1981), with the exception of R1 and R2, which use $\sigma$ vertical coordinates. The number of vertical levels ranges from 28 (R1, R2, and 20CR)

to 91 (ERA-20C), and top levels range from 3 hPa (R1 and R2) to 0.01 hPa (MERRA, MERRA-2, and ERA-20C). Figure 3 shows approximate vertical resolutions for the reanalysis systems in log-pressure altitude, assuming a scale height of 7 km and a surface pressure of 1000 hPa. A number of key differences are evident, including large discrepancies in the height of the top level (Fig. 3a) and variations in vertical resolution through the upper troposphere and lower stratosphere (Fig. 3b).

These model grids differ from the isobaric levels on which many reanalysis products are provided. Vertical spacing associated with an example set of these isobaric levels (corresponding to ERA-40 and ERA-Interim) is included in Fig. 3 for context.

In addition to differences in the location of the model top, the treatment of upper levels varies substantially across reanalysis systems. Most of the forecast models used in reanalyses implement a so-called "sponge layer", which serves to

absorb wave energy in the upper layers of the model. Sponge layers are a concession to the fact that the model atmosphere is finite, whereas the real atmosphere is unbounded at the top. The application of enhanced diffusion in a sponge layer helps to prevent unphysical reflection of wave energy at the model top that would in turn introduce unrealistic resonance in the model atmosphere (Lindzen et al., 1968). It is worth noting, however, that diabatic heating and momentum transfer associated with the absorption of wave energy by sponge layers and other simplified representations of momentum damping

(such as Rayleigh friction) may still introduce spurious behaviour in model representations of middle atmospheric dynamics (Shepherd et al., 1996; Shepherd and Shaw, 2004). Sponge layers in ERA-40 and ERA-Interim are implemented by including an additional function in the horizontal diffusion terms at pressures less than 10 hPa. This function, which varies with wavenumber and model level, acts as an effective absorber of vertically-propagating gravity waves. The sponge layer in ERA-20C also uses this approach, along with an additional first-order diffusive mesospheric sponge layer at pressures

less than 1 hPa. All three ECMWF reanalyses also apply Rayleigh friction at pressures less than 10 hPa, but the coefficient is reduced in ERA-20C relative to ERA-40 and ERA-Interim to account for the inclusion of parametrized non-orographic gravity wave drag in ERA-20C (see also section 3.1). The sponge layers in JRA-25 and JRA-55 are implemented by gradually increasing the horizontal diffusion coefficient with height at pressures less than 100 hPa. JRA-25 applies Rayleigh damping to temperature deviations from the global layer average within the top three layers of the model, while JRA-55

applies this Rayleigh damping at all pressures less than 50 hPa. MERRA and MERRA-2 increase the horizontal divergence damping coefficient in the top nine layers of the model (pressures less than ~0.24 hPa) and reducing advection on the top level to first order. CFSR applies linear Rayleigh damping at pressures less than ~2 hPa ($\sigma < 0.002$). The horizontal diffusion coefficient also increases with scale height throughout the atmosphere in CFSR. R1, R2, and 20CR do not apply any special treatment to the model upper layers; the model tops in these systems may be thought of as lids that reflect wave

energy back into the atmosphere. Additional information on the representations of horizontal diffusion and parametrized gravity wave drag is provided in section 3.1.

## 3.1 Selected physical parametrizations

Table 3 provides references for some of the physical parametrizations used in the forecast models. Many of the families of models use similar parametrizations across generations, but these are often modified and updated for use in newer systems. For example, both ERA-40 and ERA-Interim use shortwave radiation schemes based on Fouquart and Bonnel (1980) and calculate longwave radiative transfer using the RRTM (Mlawer et al., 1997), but ERA-Interim uses an updated version of the shortwave scheme and makes hourly radiation calculations rather than 3-hourly (Dee et al., 2011). ERA-20C replaces the Fouquart and Bonnel (1980) shortwave scheme with a modified version of the RRTM (Morcrette et al., 2008), and is the first ECMWF reanalysis to use the Monte-Carlo Independent Column Approximation (McICA) for representing the radiative effects of clouds. Both JRA-25 and JRA-55 use shortwave radiative transfer schemes based on Briegleb (1992), but JRA-55 uses an updated parametrization of shortwave absorption by $O_2$, $O_3$, and $CO_2$ (Freidenreich and Ramaswamy, 1999). JRA-55 also uses an updated longwave radiation model (Chou et al., 2001), which replaces the line absorption model used in JRA-25 (Goody, 1952). R1 and R2 use identical longwave radiation schemes (Fels and Schwartzkopf, 1975; Schwartzkopf and Fels, 1991), but R1 performs radiation calculations 3-hourly on a coarser linear grid while R2 performs radiation calculations hourly on the full model grid. R2 also uses a different shortwave scheme (Chou and Lee, 1996) from that used in R1 (Lacis and Hansen, 1974). Both the longwave and shortwave schemes have been replaced by a modified version of the RRTMG (Clough et al., 2005) in CFSR/CFSv2 and 20CR. The McICA approach for representing cloud radiative effects has been implemented in CFSv2, but has not been used in CFSR or 20CR. MERRA and MERRA-2 both use the CLIRAD shortwave and longwave radiation schemes (Chou and Suarez, 1999; Chou et al., 2001).

Cloud parametrizations in the ECMWF family of reanalyses follow Tiedtke (1989) for convective clouds and Tiedtke (1993) for non-convective clouds, but with substantial differences among the three reanalyses. For example, modifications to the convective parametrization between ERA-40 and ERA-Interim yielded improvements in the diurnal cycle of convection, increases in precipitation efficiency, and the capability to distinguish shallow, mid-level, and deep convective clouds (Dee et al., 2011). ERA-Interim modified the non-convective cloud parametrization to include supersaturation with respect to ice (Tompkins et al., 2007), resulting in substantial changes in the water budget of the upper troposphere. This scheme has been further modified for ERA-20C to permit separate estimates of liquid and ice water in non-convective clouds, resulting in a more physically realistic representation of mixed-phase clouds. JRA-25 and JRA-55 both use variations of the same prognostic mass-flux type Arakawa–Schubert convection scheme (Arakawa and Schubert, 1974), but JRA-55 implements a new triggering mechanism (Xie and Zhang, 2000). MERRA-2 uses the same cloud schemes as MERRA (Table 3), but has a new set of total water probability density functions as in Molod (2012). The representation of deep convection in R1, R2, CFSR and 20CR follows Arakawa and Schubert (1974), while the representation of shallow convection follows Tiedtke (1983). The versions of these parametrizations used in CFSR and 20CR have been updated relative to those used in R1 and R2, including the addition of convective momentum mixing to the deep convection scheme and modifications to the shallow convection scheme that improve the representation of marine stratocumulus (Moorthi et al.,

2010; Saha et al., 2010). R1 and R2 both use simple empirical relationships to diagnose non-convective cloud cover from grid-scale relative humidity, but these relationships are slightly different between the two systems. CFSR and 20CR replace these empirical relationships with a simple cloud physics parametrization with prognostic cloud condensate (Xu and Randall, 1996; Zhao and Carr, 1997).

Gravity wave drag and ozone parametrizations are of particular interest to the SPARC community. For example, the details of the parametrization of gravity wave drag (and particularly non-orographic gravity wave drag) can greatly influence the simulation of the QBO. All of the reanalysis systems include representations of orographic gravity wave sources and drag, but only MERRA, MERRA-2, CFSv2, and ERA-20C include parametrizations of non-orographic gravity wave drag. MERRA-2 uses a modified version of the gravity wave drag schemes used in MERRA (McFarlane, 1987; Garcia and

Boville, 1994), with enhanced intermittency and a larger non-orographic gravity wave background source in the tropics (Molod et al., 2015). The GEOS-5 forecast model does not produce a QBO before these changes are implemented, but does produce a QBO afterward. Starting with the September 2009 version (Cycle 35r3), the ECMWF IFS includes the non-orographic gravity wave drag parametrization proposed by Scinocca (2003). The version of the IFS model used for ERA-20C (the first ECMWF reanalysis to include this parametrization) produces a QBO (Hersbach et al., 2015), but with a

shorter period than observed and a weak semi-annual oscillation (SAO). CFSv2 includes a non-orographic gravity wave parametrization that considers stationary gravity waves generated by deep convection (Chun and Baik, 1998; Saha et al., 2014); this parametrization was not included in CFSR. Despite the inclusion of this parametrization, the CFSv2 forecast model does not produce a QBO. The ozone parametrizations used in current reanalysis systems are introduced and discussed in Section 6.1.

Forecast model representations of horizontal and vertical diffusion have strong influences on tracer transport and thermodynamic structure, particularly near the tropopause (Flannaghan and Fueglistaler, 2011, 2014). All of the models using spectral dynamical cores include implicit linear horizontal diffusion. These implicit representations are second order (R1, R2, and 20CR), fourth order (ERA-40, ERA-Interim, ERA-20C, JRA-25, and JRA-55), or eighth order (CFSR) in spectral space. Horizontal diffusion along model sigma layers in R1 causes the occurrence of spurious "spectral

precipitation", particularly in mountainous areas at high latitudes (Kanamitsu et al., 2002). A special precipitation product was produced for R1 to address this issue, which is greatly reduced in R2. The finite volume dynamical cores used for MERRA and MERRA-2 do not include implicit diffusion, so an explicit formulation is required. Both MERRA and MERRA-2 include explicit second-order horizontal divergence damping with a dimensionless coefficient of 0.0075 below the sponge layer. MERRA-2 also includes a second-order Smagorinsky divergence damping with a dimensionless

coefficient of 0.2 that was not applied in MERRA. The approaches to horizontal diffusion used in reanalysis schemes have been discussed in detail by Jablonowski and Williamson (2011). Approaches to vertical diffusion in the free atmosphere (above the boundary layer) are all based on the local Richardson number. ERA-40, ERA-Interim, and ERA-20C use the revised Louis scheme (Louis, 1979; Beljaars, 1995; Flannaghan and Fueglistaler, 2011), while JRA-25 and JRA-55 use the level 2 turbulence closure proposed by Mellor and Yamada (1974). R1, R2, CFSR, and 20CR use the local $K$ closure

proposed by Louis et al. (1982) in the free troposphere, but with different specifications of the background diffusion coefficients. Background diffusion coefficients in R1 and R2 are uniform throughout the atmosphere, which results in very strong vertical mixing across the tropical tropopause (Wright and Fueglistaler, 2013). By contrast, background diffusion coefficients in CFSR and 20CR decay exponentially with height from a surface value of 1 $m^2\ s^{-1}$ (Saha et al., 2010). Free

tropospheric vertical diffusion in MERRA and MERRA-2 is also parametrized based on the Louis et al. (1982) scheme, with diffusion coefficients based on the gradient Richardson number; however, a tuning parameter applied in MERRA severely suppressed turbulent mixing at pressures less than ~900 hPa. That restriction has been removed in MERRA-2, but diffusion coefficients are still usually quite small in the free atmosphere. Parametrizations of vertical diffusion within the boundary layer vary more widely amongst reanalysis systems. These parametrizations are documented in Chapter 2 of the planned S-

RIP reports, but we do not discuss them here.

### 3.2 Boundary conditions

Boundary and other specified conditions may be regarded as "externally supplied forcings" to the forecast model. These conditions include all elements of the reanalysis system that are not taken from the forecast model or data assimilation but are used to produce the outputs. The factors that can be considered "external" vary somewhat among reanalyses because the

forecast and assimilation components have provided a progressively larger fraction of the inputs for the forecast model as reanalysis systems have increased in sophistication. For example, while most of the reanalyses are run with specified SSTs and sea ice concentrations, CFSR and CFSv2 are coupled atmosphere–ocean–sea ice reanalysis systems. SST and sea ice lower boundary conditions for the CFSR and CFSv2 atmospheric models are therefore generated by coupled ocean and sea ice models (although temperatures at the atmosphere–ocean interface are relaxed every six hours to separate SST analyses

like those used by other reanalysis systems; Saha et al., 2010). Table 4 lists the SST and sea ice analyses used by the reanalysis systems. Several reanalyses use different SSTs and sea ice concentrations for different time periods, which can lead to temporal discontinuities in reanalysis products (e.g., Simmons et al., 2010). Bosilovich et al. (2015; their section 8a) further discuss these discontinuities and the steps taken in MERRA-2 to limit them, and provide a cursory graphical intercomparison of the SST fields used in the production of several recent reanalyses. Ozone is another prime example of a

quantity that may either be internally generated or externally imposed, with particular relevance to SPARC studies. The treatment of ozone in these reanalysis systems is discussed in Section 6.1.

Treatment of aerosols and other trace gases also differs: MERRA-2 assimilates aerosol optical depths and uses these analysed aerosol fields for radiation calculations (Randles et al., 2016), while most other systems use climatological aerosol fields. Different aerosol climatologies are used in ERA-40 (Tanré et al., 1984), ERA-Interim (Tegen et al., 1997), JRA-25

and JRA-55 (WMO, 1986), MERRA (Colarco et al., 2010), and CFSR and 20CR (Koepke et al., 1997). ERA-20C uses decadally-varying monthly aerosol fields prepared for CMIP5 (Lamarque et al., 2010; van Vuuren et al., 2011; Hersbach et al., 2015), while R1 and R2 neglect the role of aerosols altogether. Of the systems using prescribed aerosol fields, only CFSR, 20CR, and ERA-20C adjust them to account for the effects of volcanic eruptions. Therefore, in the majority of

reanalyses, the volcanic response in many dynamical and chemical variables is entirely due to the influences of assimilated observations. MERRA-2 aerosol analyses, which are produced using the GOCART model (Chin et al., 2002) and the Goddard Aerosol Assimilation System (Buchard et al., 2015; Randles et al., 2016), track the evolution of black and organic carbon, dust, sea salt, and sulfates. These analyses are supported by the assimilation of bias-corrected AOD at 550 nm from a variety of remote sensing platforms, including AVHRR (1980–2000), MODIS instruments on the Terra (2000–present) and Aqua (2002–present) satellites, MISR over bright surfaces (2000–2014), and the ground-based AERONET (1999–2014). Analysed aerosols, including volcanic aerosols, interact with the MERRA-2 meteorological state via direct radiative coupling.

The assumptions governing greenhouse gas concentrations also vary widely. For example, the treatment of $CO_2$ ranges from assumptions of constant global mean $CO_2$ (330 ppmv in R1; 350 ppmv in R2; 375 ppmv in JRA-25) to a linear trend extrapolated from observed 1990 values (ERA-40 and ERA-Interim), to various permutations of historical observations and future emissions scenarios (all other systems). Climatological values of several other trace gases, including $CH_4$, $N_2O$, major chlorofluorocarbons (CFCs), and occasionally hydrochlorofluorocarbons (HCFCs), are used in most of the reanalysis systems but are not included in R1, R2, and JRA-25. Radiatively active trace gases (including $CO_2$) are generally assumed to be globally well-mixed within the atmosphere, with a few notable exceptions. First, ERA-20C applies a rescaling to CMIP-5 recommended values that varies by latitude, height, month, and species (see Hersbach et al., 2015, for details). Second, distributions of $CH_4$, $N_2O$, CFCs, and HCFCs used in MERRA and MERRA-2 are based on steady-state monthly mean climatologies generated using a two-dimensional chemistry transport model; these climatologies vary by latitude, height, and month, but trace gas concentrations do not change from year to year. Third, CFSR and 20CR (after 1955) use monthly $15°\times15°$ gridded estimates of $CO_2$ derived from historical WMO Global Atmospheric Watch observations. Figure 4a–b shows temporal variations in prescribed values of $CO_2$ and $CH_4$. For simplicity, the base values of $CO_2$ and $CH_4$ (before rescaling) are shown for ERA-20C, while values of $CH_4$ for MERRA and MERRA-2 have been calculated using a mass- and area-weighted integral between 1000 and 288 hPa (seasonal variability in tropospheric methane is small in this climatology, and is therefore omitted in the integrated estimate shown in Fig. 4b).

The representation of solar radiation at the top of the atmosphere (TOA) also varies by reanalysis. Most reanalyses assume a constant total solar irradiance (TSI) of 1365 W m$^{-2}$ (R2, JRA-25, JRA-55, and MERRA), 1367.4 W m$^{-2}$ (R1), or 1370 W m$^{-2}$ (ERA-40 and ERA-Interim). These reanalyses therefore do not explicitly account for the ~11-year solar cycle in the radiative calculations, although the influences of this cycle may be introduced into the reanalysis via the assimilated observations (or in some cases via boundary conditions; see also Simmons et al., 2014). MERRA-2 and ERA-20C use TSI variations provided for CMIP5 historical simulations by the SPARC SOLARIS working group (Lean, 2000; Wang et al., 2005), with the Total Irradiance Monitor (TIM) correction applied. These variations account for solar cycle changes through mid-2008 and repeat the final cycle (April 1996–June 2008) thereafter, with magnitudes ranging from 1360.2 to 1362.7 W m$^{-2}$ between 1900 and 2008 and from 1360.6 to 1362.5 W m$^{-2}$ between 1980 and 2008. CFSR and 20CR use annual average TSI variations ranging from 1365.7 to 1367.0 W m$^{-2}$ based on data prepared by Huug van den Dool (personal

communication, 2006). The solar cycle before 1944 is repeated backwards for 20CR (e.g. insolation for 1943 is the same as that for 1954, that for 1942 is the same as that for 1953, and so on) and the solar cycle after 2006 is repeated forwards in a similar manner for both CFSR and 20CR. A programming error in ERA-40 and ERA-Interim artificially increased the effective TSI by about 2 W m$^{-2}$ relative to the specified value (so that the effective TSI is ~1372 W m$^{-2}$ rather than 1370 W m$^{-2}$). Dee et al. (2011) reported that the impact of this error is mainly expressed as a warm bias of approximately 1 K in the upper stratosphere; systematic errors in other regions are negligible. Figure 4c shows temporal variations in prescribed values of TSI from 1979 through 2015. To better highlight key differences among the reanalyses, seasonal variations resulting from the eccentricity of Earth's orbit around the sun are omitted from the figure. These seasonal variations have peak-to-peak amplitudes of ~6.8%, approximately one order of magnitude larger than the maximum difference among TSI estimates shown in Fig. 4c.

## 4 Data assimilation

This section provides a cursory overview of data assimilation concepts and methods as implemented in current reanalysis systems. Detailed summaries of data assimilation and its logical and mathematical foundations have been provided by Lorenc (1986), Daley (1993), Krishnamurti and Bounoua (1996), Bouttier and Courtier (1999), Kalnay (2003), Evensen (2009), and Nichols (2010), among others. As discussed above, an analysis is a best estimate of the state of a system, in this case the Earth's atmosphere. Data ingested into an atmospheric analysis system include observations and variables from a first guess background state (such as a previous analysis or forecast). Both the observations and the background state include important information, and neither should be considered 'truth' (as both include errors and uncertainties). An effective analysis system reduces (on balance) the errors and uncertainties associated with both observations and the first-guess background state, and therefore requires consistent and objective strategies for minimizing differences between the analysis and the (unknown) true state of the atmosphere. Such strategies often employ statistics to represent the range of potential uncertainties in the background state, observations, and techniques used to convert between model and observational space (such as spatial interpolation techniques or vertical weighting functions). Analysis systems are also generally constructed to ensure consistency with known or assumed physical properties (such as smoothness, hydrostatic balance, geostrophic or gradient-flow balance, or more complex non-linear balances). Ensembles of analyses may be used to generate useful estimates of the uncertainties in the analysis state.

The analysis methods used by current reanalysis systems include variational methods (3D-Var and 4D-Var) and the ensemble Kalman filter (EnKF). Variational methods (e.g. Talagrand, 2010) minimize a cost function that penalizes differences between observations and the model background state, with consideration of associated uncertainties. Implementations of variational data assimilation may be applied to derive optimal states at discrete times (3D-Var), or to identify optimal state trajectories within finite time windows (4D-Var). In EnKF (e.g. Evensen, 2009), an ensemble of forecasts is used to define a probability distribution of background states (the prior distribution), which is then combined with observations (and associated uncertainties) to derive a probability distribution of analysis states (the posterior

distribution). The optimal analysis state is determined by applying a Kalman filter (Kalman, 1960) to this posterior distribution (see also Evensen and van Leeuwen, 2000). One of the key advantages of 3D-Var, 4D-Var, and EnKF methods relative to many earlier implementations of data assimilation is the ability to account for nonlinear relationships between observed quantities and analysis variables. This ability to use nonlinear observation operators permits the direct assimilation of satellite radiance data without an intermediate retrieval step (Tsuyuki and Miyoshi, 2007), and underpins many of the recent advances in reanalysis development.

Figure 5 shows simplified one-dimensional schematic representations of four data assimilation strategies used in current reanalysis systems (3D-Var, 3D-FGAT, 4D-Var, and EnKF). In the following discussion, we frequently refer to the assimilation increment, which is defined as the adjustment applied to the first guess (forecast) background state following the assimilation of observational data (i.e. the difference between the analysis state and the first guess background state). We also use the term observation increment, which refers to the weighted contribution of a specific observation to the assimilation increment. The assimilation increment therefore reflects the combination of all observation increments within an assimilation window, where the latter is the time period containing observations that influence the analysis. The assimilation window used in reanalyses is typically between 6 and 12 hours long, and is often (but not always) centred at the analysis time. Core differences among the data assimilation strategies used in current reanalysis systems can be understood in terms of how the assimilation increment is calculated and applied.

The 3D-Var method (Fig. 5a) calculates and applies assimilation increments only at discrete analysis times. Observation increments within the assimilation window may either be treated as though they were all at the analysis time (which approximates the average observation time) or weighted by when they occurred (so that observations collected closer to the analysis time have a stronger impact on the assimilation increment). JRA-25 uses a 3D-Var method for data assimilation under the former assumption, in which all observations within the assimilation window are treated as valid at the analysis time. In practice, many 3D-Var systems estimate observation increments at observation times rather than analysis times (Fig. 5b). This approach is referred to as 3D-FGAT ("first guess at the appropriate time"; Lawless, 2010). The implementation of 3D-FGAT in reanalysis systems varies. For example, R1 and R2 estimate observation increments using a linear interpolation between the initial and final states of the forecast before the analysis time and a constant extrapolated value after the analysis time (this is the approach illustrated in Fig. 5b). Other 3D-FGAT systems break each forecast into multiple piecewise segments of 30 minutes (ERA-40), one hour (CFSR), or three hours (MERRA and MERRA-2) in length. The observation increments are then calculated by interpolating to observation times within each piecewise segment.

MERRA and MERRA-2 include an additional step relative to other 3D-FGAT systems, and generate two separate sets of reanalysis products (designated "ANA" for analysis state and "ASM" for assimilated state) using an iterative predictor–corrector approach (Rienecker et al., 2011). The "ANA" products are analogous to the analyses produced by other 3D-FGAT systems, and are generated by using the data assimilation scheme to adjust the background state produced by a 12-h "predictor" forecast (from 9 h before the analysis time to 3 h after). The "ASM" products, which have no analogue among other 3D-FGAT reanalyses, are generated by conducting a 6-h "corrector" forecast centred on the analysis time and using

incremental analysis update (IAU; Bloom, 1996) to apply the previously calculated assimilation increment gradually (at 30 minute intervals) rather than abruptly at the analysis time. The corrector forecast thus generates a more complete suite of atmospheric variables and tendency terms (the "ASM" products) that remains consistent with the assimilation increment while reducing wind and tracer imbalances relative to the 3D-FGAT analysis. The corrector forecast is then extended 6 h to generate the next predictor state.

Unlike 3D-Var and 3D-FGAT, which optimize the fit between assimilated observations and the atmospheric state at discrete analysis times, 4D-Var (Fig. 5c) optimizes the fit between assimilated observations and the time-varying forecast trajectory within the full assimilation window (e.g. Park and Županski, 2003). 4D-Var makes more complete use of observations collected between analysis times than 3D-Var or 3D-FGAT, and has been shown to substantially improve the resulting analysis (Talagrand, 2010). However, the computational resources required to run a 4D-Var analysis are much greater than the computational resources required to run a 3D-Var or 3D-FGAT analysis, and the full implementation of 4D-Var remains impractical at present. Current reanalysis systems using 4D-Var (e.g. ERA-Interim, ERA-20C, and JRA-55) therefore apply the simplified "incremental 4D-Var" approach described by Courtier et al. (1994). Under this approach, the model state at the beginning of the assimilation window is iteratively adjusted to obtain progressively better fits between the assimilated observations and the forecast trajectory. This iterative adjustment process propagates information both forward and backward in time, which substantially benefits the analysis but requires the derivation and maintenance of an adjoint model. The latter is a difficult and time-consuming process, and is a significant impediment to the implementation of 4D-Var. Incremental 4D-Var is tractable (unlike full 4D-Var), but it is still computationally expensive, and is therefore usually implemented in two nested loops for computational efficiency. Assimilation increments are first tested and refined in an inner loop with reduced resolution and simplified physics, and then applied in an outer loop with full resolution and full physics after the inner loop converges.

Most implementations of variational methods in reanalyses systems are based on single deterministic forecasts. By contrast, EnKF (Fig. 5d) uses an ensemble approach to evaluate and apply assimilation increments. Major advantages of the ensemble Kalman filter technique include ease of implementation (unlike 4D-Var, EnKF does not require an adjoint model) and the generation of useful estimates of analysis uncertainties, which are difficult to obtain when using variational techniques with single forecasts (the forthcoming ERA5 will use 4D-Var in an ensemble framework, in part to address this issue). Whitaker et al. (2009) found that in the case of a reanalysis that assimilates only surface pressure observations, the performance of the 4D-Var and EnKF techniques is comparable, and that both 4D-Var and EnKF give more accurate results than 3D-Var. 20CR uses an EnKF method for data assimilation.

Additional details regarding these data assimilation methods, including a fuller discussion of the relative advantages and disadvantages, are beyond the scope of this paper. These issues have been discussed and summarized by Park and Županski (2003), Lorenc and Rawlins (2005), Kalnay et al. (2007a; 2007b), Gustafsson (2007), and Buehner et al. (2010a; 2010b), among others.

## 5 Input observations

Reanalysis systems assimilate data from a variety of sources. These sources are often grouped into two main categories: conventional data (e.g. surface records, radiosonde profiles, and aircraft measurements) and satellite data (e.g. microwave and infrared radiances, atmospheric motion vectors inferred from satellite imagery, and various retrieved quantities). The density and distribution of these data have changed considerably over time. Conventional data are unevenly distributed in space and time. Satellite data are often more evenly distributed in space but still inhomogeneous, and the availability of these data has changed over time as sensors have been introduced and retired. Both types of data have generally become denser over time. Such changes in the availability of input observations have strong impacts on the quality of the reanalyses that assimilate them, so that discontinuities in reanalysis data should be carefully evaluated and checked for coincidence with changes in the input observing systems. The quality of a given type of measurement is also not necessarily uniform in time. For example, virtually all radiosonde sites have adopted new instrument packages at various times, while TOVS and ATOVS satellite data were collected using several different sounders on several different satellites with availability and biases that changed substantially over time. Almost all observing systems suffer from biases that must be corrected before the data can be assimilated, as well as jumps and drifts in the time series that cause the quality of reanalysis products to change over time. Bias corrections prior to and/or within the assimilation step are therefore essential for creating reliable reanalysis products.

Although modern reanalysis systems assimilate observations from many common sources, different reanalysis systems assimilate different subsets of the available observations. Such discrepancies are particularly pronounced for certain categories of satellite observations and, like differences in the underlying forecast models, are an important potential source of inter-reanalysis differences. Moreover, the assimilation of observational data contributes directly to deficiencies in how reanalyses represent the state and variability of the upper troposphere, stratosphere, and mesosphere. For example, data assimilation can act to smooth sharp vertical gradients in the vicinity of the tropopause. The potential importance of this effect is illustrated by abrupt changes in vertical stratification near the tropopause at the beginning of the satellite era in R1 (Birner et al., 2006). Changes in data sources and availability can also lead to biases and artificial oscillations in temperature in various regions of the stratosphere, particularly in the polar and upper stratosphere where observations are sparse (Randel et al., 2004; Uppala et al., 2005; Simmons et al., 2014; Lawrence et al., 2015). Information and errors introduced by the input data and data assimilation system propagate upwards through the middle atmosphere in both resolved waves and parametrized gravity wave drag (Polavarapu and Pulido, 2016). The abrupt application of assimilation increments can generate spurious gravity waves in systems that use intermittent data assimilation techniques (Schoeberl et al., 2003), including most implementations of 3D-Var, 3D-FGAT, and EnKF, and may also generate instabilities that artificially enhance mixing and transport in the subtropical lower stratosphere (Tan et al., 2004). Reanalyses of the stratosphere and mesosphere are therefore sensitive not only to the model formulation and input data at those levels, but also to the details of the data assimilation scheme and input data at lower altitudes.

## 5.1 Conventional data

Radiosondes provide high vertical resolution profiles of temperature, horizontal wind, and humidity worldwide, although most radiosonde stations are located in the Northern Hemisphere at middle and high latitudes over land. The typical vertical coverage of radiosonde data extends from the surface up to 30~10 hPa for temperature and winds and from the surface up to 300~200 hPa for humidity. The main source of systematic errors in radiosonde temperature measurements stems from the effects of solar radiative heating and (to a lesser extent) infrared cooling on the temperature sensor (Nash et al., 2011). This issue, which is sometimes called the "radiation error", can cause pronounced warm biases in raw daytime stratospheric measurements. These biases may be corrected onsite in the ground data receiving system before reporting, and further corrections may be applied at each reanalysis centre before assimilation. The major issue with radiosonde humidity measurements is that the sensor response has historically been too slow at low temperatures (Nash et al., 2011). Radiosonde observations of humidity at pressures less than about 300 hPa are therefore often unreported and/or excluded from the assimilation. Recent advances in radiosonde instrumentation are beginning to improve this situation, although operational radiosondes remain unable to provide accurate estimates of humidity in the stratosphere. Other issues include frequent (and often undocumented) changes in radiosonde instrumentation and observing methods at radiosonde stations, which may cause jumps in the time series of temperature, relative humidity, and winds. Several "homogenization" activities for radiosonde temperature data exist to support climate monitoring and trend analyses, in which observations from different launch sites and instrument suites are post-processed to remove biases, drifts, and jumps in the data record. Although some of these activities have been conducted independently of reanalysis activities (e.g. Sherwood, 2007), others (notably RAOBCORE; Haimberger et al., 2008, 2012) have been conducted with reanalysis applications in mind. One or more versions of RAOBCORE are used in ERA-Interim (v1.3), MERRA and MERRA-2 (v1.4 through 2005), and JRA-55 (v1.4 through 2005; v1.5 thereafter). Further efforts on data rescue, reprocessing, homogenization, and uncertainty evaluation by the broader research community are likely to be an essential part of the next generation of reanalyses (e.g., ACRE and GRUAN; Allan et al., 2011; Bodeker et al., 2016).

Figure 6 shows timelines of conventional data sources assimilated by ERA-Interim, JRA-55, MERRA, MERRA-2, and CFSR. These timelines are quite consistent among modern full input reanalyses (as well as the conventional input JRA-55C). All of the reanalysis systems listed in Table 1 assimilate records of surface pressure from manned and automated weather stations, ships, and buoys, while all but 20CR assimilate at least some records of surface winds over oceans. All but ERA-Interim, ERA-20C, 20CR, and JRA-55C assimilated synthetic surface pressure data for the Southern Hemisphere (PAOBS). PAOBS are subjective analyses of surface pressure produced by the Australian Bureau of Meteorology based on available observations and temporal continuity, which are used to compensate for the scarcity of direct observations in the Southern Hemisphere. The influence of these data in reanalysis systems has waned in recent years, as the availability of direct observations covering the Southern Hemisphere has expanded. All of the full input reanalyses and JRA-55C assimilate upper-air observations made by radiosondes, dropsondes, and wind profilers. JRA-25, JRA-55, and JRA-55C assimilate

wind speed profiles in tropical cyclones, while 20CR assimilates records of tropical cyclone central pressures. CFSR uses the NCEP tropical storm relocation package (Liu et al., 1999) to relocate tropical storm vortices to observed locations.

Although different reanalysis systems may assimilate (or exclude via quality control) different subsets of conventional data within the broader categories shown in Fig. 6, conventional data are often shared among reanalysis centres. Major quality control criteria for conventional data include checks for completeness, physical and climatological consistency, and duplicate reports. Data may also be filtered using locally compiled blacklists or blacklists acquired from other data providers and reanalysis centres. Detailed intercomparisons of the conventional data or quality control criteria used in each reanalysis are beyond the scope of this review; however, four of the reanalyses that assimilate upper-air observations (ERA-40, ERA-Interim, JRA-25, JRA-55, and JRA-55C) use the ERA-40 ingest as a starting point, and the ERA-40 ingest has much in common with the conventional data archives used by NCEP (R1, R2, and CFSR) and the NASA GMAO (MERRA and MERRA-2). More recent updates in data holdings at ECMWF, JMA, GMAO, and NCEP rely heavily on near-real-time data gathered from the WMO Global Telecommunication System (GTS), which also contributes to the use of a largely (but not completely) common set of conventional data among reanalysis systems.

Measurements made by aircraft, such as the Aircraft Meteorological Data Relay (AMDAR) collection, are influential inputs in many atmospheric analyses and reanalyses (Petersen, 2016). Horizontal wind data from aircraft are assimilated in all of the reanalysis systems except JRA-55C, ERA-20C, and 20CR, while temperature data from aircraft are assimilated in all of the reanalysis systems except JRA-25, JRA-55 (and JRA-55C), ERA-20C, and 20CR. In principle, aircraft data were assimilated from the outset by ERA-40 (September 1957; Uppala et al., 2005), JRA-55 (January 1958; Kobayashi et al., 2015), and R1 (January 1958; Kalnay et al., 1996), although many of the data from these early years do not meet the necessary standards for assimilation. The volume of aircraft data suitable for assimilation increased substantially after January 1973 (Uppala et al., 2005; Kobayashi et al., 2015). Aircraft temperature data have been reported to have a warm bias with respect to radiosonde observations (Ballish and Kumar, 2008). This type of discrepancy among ingested data sources can have important impacts on the analysis. For example, Rienecker et al. (2011) and Simmons et al. (2014) have shown that an increase in the magnitude of the temperature bias at 300 hPa in MERRA with respect to radiosondes in the middle to late 1990s coincides with a large increase in the number of aircraft observations assimilated by the system, and that differences in temperature trends at 200 hPa between MERRA and ERA-Interim reflect the different impacts of aircraft temperatures in these two reanalysis systems. MERRA-2 applies adaptive bias corrections to AMDAR observations that may help to reduce the uncertainties associated with assimilating these data (Bosilovich et al., 2015): after each analysis step the updated bias is estimated as a weighted running mean of the aircraft observation increments from preceding analysis times. These adaptive bias corrections are calculated and applied for each aircraft tail number in the database separately.

## 5.2 Satellite radiances

Operational satellite radiance measurements provide constraints for temperature and moisture with more homogeneous spatial coverage than radiosondes, but with coarser vertical resolutions and deep vertical weighting functions (Fig. 7). All of

the full input reanalyses described in this paper (i.e. excluding the conventional input reanalysis JRA-55C and the surface input reanalyses ERA-20C and 20CR) assimilate some form of satellite measurements after the coverage of those measurements expanded in the 1970s. The earliest satellite data assimilated by these systems are VTPR radiances, which were assimilated by ERA-40 and JRA-55 from January 1973 through late 1978 (ERA-40) or early 1979 (JRA-55). The most prevalent satellite data assimilated by reanalyses are observations made by microwave and infrared sounders in the TOVS suite (October 1978–October 2006 on multiple satellites) and the ATOVS suite (October 1998 to present on multiple satellites), which influence the global temperature and moisture analyses produced by all of the full input reanalysis systems described in this paper. The TOVS suite included SSU, MSU, and HIRS. The ATOVS suite includes the AMSU-A, AMSU-B, and more recent versions of HIRS. R1 and R2 assimilate temperature retrievals from these instruments (e.g. Reale, 2001), while the other full input reanalysis systems assimilated radiance data directly. The assimilation of satellite radiances requires the use of a radiative transfer scheme, which differs from the radiative transfer scheme used in the forecast model. Several systems use one or more versions of RTTOV, including RTTOV-5 (ERA-40), RTTOV-6 (TOVS radiances in JRA-25), RTTOV-7 (ERA-Interim and ATOVS radiances in JRA-25), and RTTOV-9 (JRA-55). MERRA uses the GLATOVS model for assimilating SSU radiances and the CRTM for assimilating all other radiances. MERRA-2 and CFSR use the CRTM for all radiances.

Figure 8 shows timelines of satellite radiances assimilated by ERA-Interim, JRA-55, MERRA, MERRA-2, and CFSR. In addition to the TOVS and ATOVS suites, several reanalysis systems also assimilate radiances from AIRS (ERA-Interim, MERRA, MERRA-2, and CFSR). MERRA-2 and CFSR assimilate hyperspectral radiances from IASI, while MERRA-2 also assimilates radiances from the hyperspectral sounder CrIS and the most recent generation of microwave sounder ATMS.

Raw radiance data often include drifts and jumps due to orbital drift, inaccurate calibration offsets, long-term trends in atmospheric $CO_2$, and other issues (Zou et al., 2006, 2014; Simmons et al., 2014). Moreover, the overlap periods between successive instruments are often short, which complicates efforts to adjust for these issues. The biases associated with these drifts and jumps can propagate into the reanalysis fields. For example, Rienecker et al. (2011) speculated that artificial annual cycles emerge in upper stratospheric temperature in MERRA because variations in atmospheric $CO_2$ are not considered in the GLATOVS radiative transfer model used to assimilate SSU radiances (these issues have been corrected in MERRA-2, which uses the CRTM to assimilate SSU radiances). Discontinuities associated with the TOVS-to-ATOVS transition show up in several aspects of the reanalysis, but are particularly pronounced in the upper stratosphere and lower mesosphere (e.g. Simmons et al., 2014). The strong influence of the TOVS-to-ATOVS transition in the middle atmosphere is mainly attributable to the substantial improvement in vertical resolution involved in switching from SSU (Fig. 7a) to AMSU-A (Fig. 7b). A further example of the influence of ATOVS is provided by the cold bias (~2 K) in middle stratospheric temperatures that persisted in JRA-25 between 1979 and 1998 (Onogi et al., 2007). This feature resulted from a known cold bias in the radiative transfer model used by JRA-25. SSU had only three channels sensitive to stratospheric temperature, too few to correct the model bias, whereas assimilation of the higher-resolution AMSU-A radiances starting in

1998 was sufficient to effectively correct the model bias. Discontinuities associated with the TOVS-to-ATOVS transition will be discussed in more detail in the planned S-RIP reports.

Post-launch inter-satellite calibration (or "homogenization") efforts by the satellite remote sensing community (such as the WMO GSICS; Goldberg et al., 2011) have substantially reduced inter-satellite differences in several cases, including MSU (Zou et al., 2006), AMSU-A (Zou and Wang, 2011), and SSU (Zou et al., 2014). This type of inter-satellite calibration has historically been performed by reanalysis systems internally via bias correction terms applied during the data assimilation step. In older reanalyses that assimilated satellite radiances, such as ERA-40 and JRA-25, bias corrections were often (but not always) based on a fixed regression that spanned the lifetime of the instrument (Uppala et al., 2005; Onogi et al., 2007; Sakamoto and Christy, 2009). This approach, which occasionally required the reanalysis to be interrupted for manual retuning of bias correction terms, has been replaced by adaptive (or variational) bias correction schemes in recent reanalysis systems. Adaptive bias corrections for satellite radiances are based on differences between observed radiances and expected radiances calculated from model-generated background states. Some early implementations of adaptive bias corrections, such as that applied to TOVS data in JRA-25, left the reanalysis vulnerable to jumps and drifts inherited from the assimilated radiances (Sakamoto and Christy, 2009). These problems are addressed in most recent reanalysis systems by defining observational "anchors" that are regarded as unbiased and are therefore allowed to contribute directly to the background state (Dee, 2005). A key example is the use of homogenized radiosonde data (Section 5.1) to anchor bias corrections for satellite radiances (e.g. Auligné et al., 2007). Versions of this approach have been implemented in ERA-Interim, JRA-55, MERRA, and MERRA-2. Global Navigation Satellite System radio occultation (GNSS-RO) observations (Section 5.3) are also useful for anchoring bias corrections (e.g. Poli et al., 2010), and have been used in this capacity in ERA-Interim, JRA-55, and MERRA-2; however, GNSS-RO data are only available after May 2001. The approach used in CFSR and CFSv2 (Derber and Wu, 1998; Saha et al., 2010) differs, in that anchor observations are not used to adjust the background state prior to assimilation of satellite radiances. Instead, initial bias corrections are determined for each new satellite instrument via a three-month spin-up assimilation and then allowed to evolve slowly. The effects of satellite-specific drifts and jumps are kept small by assigning very low weights to the most recent biases between the observed and expected radiances, and by accounting for known historical variations in satellite performance as catalogued by multiple research centres. It is therefore not strictly necessary for satellite data to be homogenized prior to its assimilation in a reanalysis system, although it is beneficial for the system to assimilate data with biases that are as small as possible.

The use of externally homogenized satellite radiances has been found to improve some aspects of recent reanalyses. For example, homogenized MSU data (Zou et al., 2006) assimilated by CFSR, MERRA, and MERRA-2 have been found to improve temporal consistency in bias correction patterns (Rienecker et al., 2011), and may have helped MERRA to produce a more realistic stratospheric temperature response than ERA-Interim following the eruption of Mount Pinatubo (Simmons et al., 2014). Homogenized radiance data may be even more effective in eliminating artificial drifts and jumps in the analysis state in situations where conventional data are unavailable or insufficient to provide a reference for satellite bias correction, such as SSU in the middle and upper stratosphere. Homogenized satellite radiance time series only represent a relatively

small fraction of the satellite data ingested by current reanalysis systems (many of which do not assimilate homogenized data at all); however, the availability of homogenized satellite radiance time series is increasing and these data are likely to become more influential in future reanalysis efforts.

Quality control checks for satellite radiance data vary by satellite sensor and reanalysis system, and also often differ for different spectral channels of the same sensor. Commonly used criteria include the exclusion of radiances affected by the presence of clouds or rain, and radiances measured over certain types of surfaces (e.g. land or ocean, snow/ice, high terrain). The information assimilated from a given sensor may also differ among reanalyses. For example, MERRA assimilates channels 1 through 15 from AMSU-A, while CFSR assimilates channels 1 through 13 and channel 15 (excluding channel 14), JRA-55 and MERRA-2 assimilate channels 4 through 14 (excluding channels 1 through 3 and channel 15), JRA-25 assimilates channels 4 through 13 (excluding channels 1 through 3, channel 14, and channel 15), and ERA-Interim assimilates channels 5 through 14 (excluding channels 1 through 4 and channel 15). Such differences are not unique to AMSU-A. Detailed discussion of the satellite radiance quality control criteria applied in modern reanalysis systems is beyond the scope of this paper; additional information can be found in the publications listed in Table 1.

### 5.3 Other satellite data sets

In addition to the satellite sounding data discussed above, atmospheric motion vector (AMV) data derived from geostationary and polar-orbiting satellite images have relatively large influences on reanalysis fields in the upper troposphere and lower stratosphere, as do satellite ozone retrievals and other satellite-derived quantities. Figures 9 and 10 show timelines of AMVs and other non-radiance satellite data assimilated by ERA-Interim, JRA-55, MERRA, MERRA-2, and CFSR. AMVs derived from satellite imagery are assimilated by all of the full input reanalysis systems, although the data sources and temporal coverage differ. All of the full input systems except for R1 and R2 also assimilate satellite observations of ocean surface winds (scatterometers, SSM/I, and SSMIS), but again the data sources and temporal coverage vary substantially among different systems.

ERA-Interim, JRA-55, MERRA-2, and CFSR assimilate data from GNSS-RO instruments (also referred to in the literature as GPS-RO). These data are assimilated in the form of bending angles or refractivity at the tangent point, rather than temperature or water vapour retrievals. GNSS-RO occultations are based on radio waves that are calibrated against on-board atomic clocks, and are therefore exceptionally stable both in time and across satellite platforms (Poli et al., 2010). The resulting retrievals have small random errors (equivalent to ~1 K) and very small systematic errors (less than ~0.2 K). GNSS-RO data can therefore be assimilated without bias corrections, thereby serving as 'anchors' that facilitate the derivation of more accurate bias corrections for other data sources. These data are particularly useful for constraining the vertical structure of the lower and middle stratosphere (Bonavita, 2014; Polavarapu and Pulido, 2016). MERRA-2 also assimilates temperature retrievals from Aura MLS, but only at pressures less than 5 hPa (Bosilovich et al., 2015). This choice produces discontinuities in some upper-level MERRA-2 products that coincide with the first assimilation of MLS

temperature retrievals in August 2004, but provides valuable constraints for the reanalysis in the upper stratosphere and lower mesosphere (USLM) during the Aura mission.

## 6 Ozone and water vapour

### 6.1 Ozone

All of the reanalysis systems assimilate satellite ozone measurements except for ERA-20C, R1, R2, 20CR (and JRA-55 and ERA-40 before 1978), and all except for R1 and R2 include some form of prognostic ozone parametrization and analysis. However, some of the reanalysis systems that assimilate ozone and produce ozone analyses (notably ERA-40, ERA-Interim, and ERA-20C) nevertheless use climatological ozone fields for radiation calculations in the forecast model, rather than their respective ozone analyses, because interactions between radiation and prognostic ozone have been found to

amplify errors in the analysed temperature fields (Dethof and Hólm, 2004; Dee et al., 2011). Even amongst reanalyses that use climatological fields the characteristics of these fields may differ. For example, while ERA-40 and ERA-Interim use zonal mean monthly mean fields with a repeating annual cycle (Fortuin and Langematz, 1995), ERA-20C uses monthly three-dimensional fields that evolve in time (Cionni et al., 2011).

    The prognostic ozone models used in the reanalysis systems are listed in Table 3. The ozone parametrization in ERA-40 is

15 an update of the Cariolle and Déqué (1986) scheme as described by Dethof and Hólm (2004). Recent upgrades of the scheme (Cariolle and Teyssèdre, 2007) are included in ERA-Interim (Dragani, 2011) and ERA-20C. The scheme is a linearization of the ozone continuity equation, where the linear coefficients have been computed using an external 2-D photochemical model. This 2-D model does not include heterogeneous chemistry, but the parametrization includes an ad-hoc ozone destruction term to account for the chemical loss due to polar stratospheric clouds. The ozone parametrizations in

CFSR, CFSv2, and 20CR are based on the scheme proposed by McCormack et al. (2006). This scheme also assumes linear relaxation toward a zonal mean monthly mean photochemical equilibrium state, but with a different representation of chemical loss rates and independent derivations of the reference state and model parameters. MERRA and MERRA-2 diagnose ozone using a diurnally- and height-varying empirical relationship between ozone and prognostic odd-oxygen (Rienecker et al., 2008). This partitioning only applies at pressures greater than 1 hPa; at lower pressures (higher altitudes)

all odd-oxygen is assumed to be ozone. The odd-oxygen scheme includes tracer advection and climatological monthly mean zonal mean production and loss rates, with production rates tuned for agreement with satellite-based ozone climatologies. This scheme differs from the linearized relaxation schemes discussed above in that it specifies production and loss rates directly, rather than diagnosing them as functions of the deviation from a specified reference state. JRA-25 and JRA-55 (after 1979) use a fundamentally different approach, in which daily three-dimensional ozone concentrations are estimated

using the MRI-CCM1 offline CTM (Shibata et al., 2005) and then nudged to satellite observations of total column ozone (TCO). The version of the CTM used for JRA-55 is slightly modified from that used for JRA-25, with 68 vertical levels

rather than 45. JRA-55 does not include prognostic ozone variations before 1979. The treatment of ozone in JRA-55C matches that in JRA-55 exactly both before and after 1979.

Assimilated satellite observations of ozone vary widely among reanalysis systems that produce ozone analysis products (Fig. 10). For example, JRA-25 and JRA-55 only use retrievals of TCO for nudging daily offline CTM output toward

observations, while ERA-40, ERA-Interim, MERRA, MERRA-2, and CFSR assimilate both ozone profiles and TCO. MERRA and CFSR only assimilate ozone retrievals (both TCO and profiles, with relatively coarse vertical resolution) from SBUV and SBUV/2, while ERA-40 assimilated coarse vertical resolution profiles from SBUV and TCO from TOMS starting in January 1979. MERRA-2 assimilates ozone retrievals from SBUV and SBUV/2 (both TCO and profiles) through September 2004, and assimilates OMI TCO and Aura MLS profiles thereafter. Estimates of TCO from SBUV and SBUV/2

assimilated by MERRA and MERRA-2 are calculated as the mass-weighted sum of layer values, rather than direct retrievals of TCO. ERA-Interim assimilates retrievals from a wide variety of satellite instruments, including some with relatively high vertical resolution. Ozonesonde measurements are not assimilated in current reanalysis systems, but are often used for validation.

## 6.2 Water vapour in and above the upper troposphere

The assimilation of radiosonde and satellite observations of humidity fields is problematic in the upper troposphere and above, where water vapour mixing ratios are very low. The impact of saturation means that humidity probability density functions are often highly non-Gaussian (Ingleby et al., 2013). These issues are particularly pronounced in the vicinity of the tropopause, where sharp temperature gradients complicate the calculation and application of bias corrections for humidity variables during the assimilation step. Reanalysis systems therefore often only assimilate observations of water vapour

provided by radiosondes and/or microwave and infrared sounders (usually in the form of radiances) from the surface up to a specified upper bound, which is typically between ~300 hPa and ~100 hPa. In regions of the atmosphere that lie above this upper bound (i.e. the uppermost troposphere and stratosphere), the water vapour field is typically determined by the forecast model alone. In this case, estimates of water vapour in the stratosphere are determined by some combination of transport from below, turbulent mixing, and dehydration in the vicinity of the tropical cold point tropopause (e.g. Fueglistaler et al.,

2009).

The ECMWF reanalyses considered here do not allow adjustments to the water vapour field due to data assimilation in the stratosphere (above the diagnosed tropopause), but include simple parametrizations of methane oxidation (Dethof, 2003; Monge-Sanz et al., 2013). ERA-Interim and ERA-20C also include a parametrization that allows supersaturation with respect to ice in the cloud-free portions of grid cells with temperatures less than 250 K, which yields substantial increases in

relative humidity in the upper troposphere and stratospheric polar cap in ERA-Interim relative to ERA-40 (Dee et al., 2011). JRA-25 and JRA-55 do not assimilate observations of humidity at pressures smaller than 100 hPa, and set the vertical correlations of humidity background errors to zero at pressures smaller than 50 hPa (JRA-25) or 5 hPa (JRA-55) to prevent spurious analysis increments at higher levels. JRA-25 assumes a constant mixing ratio of 2.5 ppmv in the stratosphere for

radiation calculations, while JRA-55 uses an annual mean climatology derived from HALOE and UARS MLS measurements made during 1991–1997. MERRA and MERRA-2 tightly constrain stratospheric water vapour to a specified profile, which is based on zonal mean monthly climatologies from HALOE and Aura MLS (Jiang et al., 2010; Rienecker et al., 2011). Water vapour above the tropopause does not undergo physically meaningful variations in MERRA or MERRA-2. Neither R1 nor R2 assimilates satellite humidity retrievals, and R1 does not provide analyses of moisture variables at pressures smaller than 300 hPa. CFSR only assimilates radiosonde humidities at pressures of 250 hPa and larger, although no upper altitude limit is assigned to assimilated GNSS-RO data. CFSR and 20CR provide moisture variables in the stratosphere, but dehydration processes in the tropopause layer may yield negative values. These negative values are artificially replaced by very small positive values for the radiation calculations, but are not replaced in the analysis.

In general, reanalyses do not provide physically meaningful estimates of water vapour above the tropopause, although it should be noted that observational datasets used for comparison to the models have their own rather large biases in this region (Hegglin et al., 2013). Given the importance of water vapour in the upper troposphere and lower stratosphere (UTLS) for radiative forcing (e.g., Forster and Shine, 2002; Randel et al., 2007; Gettelmann et al., 2011; Riese et al., 2012), large biases in the representation of the water vapour gradients across the tropopause and in the lower stratosphere may lead to non-negligible radiative and dynamical impacts in reanalysis systems. The magnitude of these impacts in the different reanalyses is not yet quantified, but is under investigation within S-RIP. Regardless, we emphasize in no uncertain terms that reanalysis humidity products in the upper troposphere and stratosphere should be used only with extreme caution.

## 7 Summary and outlook

In this paper we have introduced the motivation for and goals of the SPARC Reanalysis Intercomparison Project (S-RIP), and presented an overview of the reanalysis data sets that are being evaluated. Additional information on the former may be found in Chapter 1 of the planned interim and full S-RIP reports, while further details and alternative presentations of the latter may be found in Chapters 2–4 of the planned reports. These reports will be available in the SPARC report series hosted at http://www.sparc-climate.org/. We are conducting a comprehensive intercomparison of reanalyses that are extensively used to study processes in the upper troposphere and middle atmosphere. This intercomparison will provide guidance to the research community regarding which reanalyses are most suitable for various types of data analysis and modelling studies, particularly those for which dynamical and transport processes are critical. The planned S-RIP reports will be divided into individual chapters that apply process-based diagnostics from multiple reanalyses to study specific phenomena or regions of the atmosphere, and in many cases the results are presented as parts of papers focused on understanding those processes rather than purely as intercomparisons. Specific research areas covered by S-RIP include the Brewer-Dobson circulation, stratosphere–troposphere dynamical coupling, upper tropospheric/lower stratospheric processes and stratosphere–troposphere exchange in both the extratropics and tropics, the QBO and tropical variability, lower stratospheric polar chemical processing and ozone loss, and dynamics and transport in the upper stratosphere and lower

mesosphere. The two planned S-RIP reports and multiple peer-reviewed papers (including those in this special issue) will provide comprehensive documentation of the results of the S-RIP activity.

We have mentioned several aspects of reanalysis systems that can directly influence reanalysis products in the upper troposphere, stratosphere, and mesosphere. These include aspects that are present in any model-based system, such as physical parametrizations and boundary conditions (Section 3), as well as aspects that are unique to reanalyses and other systems that use data assimilation. For example, instabilities generated by intermittent data assimilation techniques (e.g. 3D-Var, 3D-FGAT, EnKF) can propagate upward and artificially enhance mixing and transport in the middle atmosphere (Schoeberl et al., 2003; Tan et al., 2004; Polavarapu and Pulido, 2016). The IAU approach used by MERRA and MERRA-2 and the incremental 4D-Var systems used by ERA-Interim, ERA-20C, and JRA-55 help to address this issue by applying assimilation increments gradually rather than all at once. The treatment of the model upper layers can also introduce artificial instabilities. Although most current reanalyses apply a diffusive 'sponge layer' in the uppermost model layers (Section 3), older reanalyses such as R1 and R2 suffer from spurious wave energy that is reflected back into the atmosphere from the model top. Sponge layers improve this situation, but can still generate artificial instabilities that modify the model behaviour (Shepherd et al., 1996; Shepherd and Shaw, 2004). The effects of such instabilities are largest in the uppermost layers of the model, where the impacts of the sponge layer are most immediate and observational constraints are sparse.

Changes in the observing system can cause drifts and jumps in reanalysis products. A particularly influential example is the change from TOVS to ATOVS that occurred in 1998 (see Section 5.2). This change causes discontinuities throughout reanalyses, but especially in the middle and upper atmosphere, where the enhanced vertical resolution of assimilated satellite radiances associated with the switch from SSU to AMSU-A has a particularly large impact (Onogi et al., 2007; Simmons et al., 2014; see also Fig. 7). Such discontinuities are not limited to the TOVS–ATOVS transition; other notable examples include the transition from NOAA-7 SSU to NOAA-9 SSU in 1985, which created evident discontinuities in the upper stratosphere in both ERA-40 and ERA-Interim, and the assimilation of Aura MLS temperature retrievals at pressures less than 5 hPa by MERRA-2, which started in August 2004. These discontinuities will be documented in detail in Chapter 3 of the planned S-RIP reports. The assimilation of homogenized satellite radiances (Section 5.2), in which observations collected by different satellites are cross-calibrated to reduce biases and eliminate discontinuities in the data record, can help to ameliorate these issues. The use of homogenized satellite radiance data in reanalyses has so far been limited to MSU (Zou et al., 2006), but even with such limited application there is evidence that assimilating these data in place of raw radiances improves temporal continuity (Rienecker et al., 2011) and helps the reanalysis to produce more realistic stratospheric temperature responses to volcanic eruptions (Simmons et al., 2014). In situations where suitable anchor observations are unavailable or insufficient to provide a reference for satellite bias correction, such as SSU in the middle and upper stratosphere, homogenized radiance data may be even more effective in eliminating artificial drifts and jumps in the analysis state. Several recent reanalysis systems also assimilate homogenized radiosonde data (Haimberger et al., 2008, 2012) to help limit the impacts of changes in the conventional observing system.

Issues can also arise from the application of bias correction procedures that are intended to limit discontinuities. For example, the evolving bias corrections used in CFSR (Saha et al., 2010; see also Section 5.2) ultimately introduce an oscillating warm bias in temperatures in the upper stratosphere. This bias, which is intrinsic to the forecast model, essentially disappears when a new execution stream is introduced (Fig. 2), only to slowly return as the model bias is imprinted on the observational bias correction terms. Discrepancies among ingested observations, such as the systematic bias between radiosonde and aircraft measurements of temperature in the upper troposphere (Ballish and Kumar, 2008), can also introduce biases in reanalysis products, particularly when the bias correction terms are fixed or poorly constrained. A large increase in upper tropospheric temperature biases in MERRA during the 1990s corresponds to a large increase in the number of assimilated aircraft observations (Rienecker et al., 2011), while discrepancies between temperature trends in this region between MERRA and ERA-Interim can be attributed to different treatments of assimilated aircraft data (Simmons et al., 2014). Improvements in algorithms for adaptive bias correction, the increasing availability of low-bias anchor observations (such as GNSS-RO bending angles, as discussed in Section 5.3), and the expanding use of ensembles for better incorporating analysis uncertainties into bias correction terms are helping to reduce the impacts of these types of errors.

In addition to supporting the use of reanalysis products in scientific studies, a fundamental goal of S-RIP is to provide well-organized feedback to the reanalysis centres, thus forming a "virtuous circle" of assessment, improvements in reanalyses, further assessment, and further improvements in reanalyses. To this end, calculations of diagnostics suited to numerous types of studies have been and are being developed for current reanalyses. These diagnostics can then be easily extended and applied to assessment of future reanalyses. By establishing tighter links between reanalysis providers and the SPARC community, outcomes from the S-RIP assessment will motivate future reanalysis developments. The initial period of the S-RIP project is about five years (through 2018); however, the tools and relationships between reanalysis providers and the community of reanalysis data users developed during the S-RIP activity will continue to facilitate both fundamental atmospheric science research and continuing progress in reanalysis development for many years after the original project has concluded.

A further legacy of S-RIP will be the creation of a public data archive of processed reanalysis data with standard formats and resolutions (see http://s-rip.ees.hokudai.ac.jp/resources/data.html). This archive will help to enable both further intercomparisons and scientific analyses without repetition of expensive pre-processing steps. The S-RIP ensemble of derived data sets will be freely available to researchers worldwide, and is intended to be a useful tool for reanalysis assessment beyond the lifetime of the project.

Although this special issue has been initiated by the S-RIP leadership, the collected papers are not exclusive to S-RIP participants, and encompass a variety of tools, issues, and results related to the intercomparison of reanalysis products throughout the atmosphere.

**Appendix A**

Major acronyms are explained below.

20CR: 20th Century Reanalysis of NOAA and CIRES

3D-FGAT: 3-dimensional variational assimilation scheme with first guess at the appropriate time

3D-Var: 3-dimensional variational assimilation scheme

4D-Var: 4-dimensional variational assimilation scheme

ACRE: Atmospheric Circulation Reconstructions over the Earth

AERONET: Aerosol Robotic Network

AIRS: Atmospheric Infrared Sounder

AMDAR: Aircraft Meteorological Data Relay

AMIP: Atmospheric Model Intercomparison Project

AMSR: Advanced Microwave Scanning Radiometer

AMSR-E: Advanced Microwave Scanning Radiometer for EOS

AMSU: Advanced Microwave Sounding Unit

AMV: atmospheric motion vectors

AOD: aerosol optical depth

Aqua: a satellite in the EOS A-Train satellite constellation

ATMS: Advanced Technology Microwave Sounder

ATOVS: Advanced TIROS Operational Vertical Sounder

Aura: a satellite in the EOS A-Train satellite constellation

AVHRR: Advanced Very High Resolution Radiometer

BUOY: surface meteorological observation report from buoys

CDAS: Climate Data Assimilation System

CFC: chlorofluorocarbon

CFSR: Climate Forecast System Reanalysis of NCEP

CFSv2: Climate Forecast System, version 2

CIRES: Cooperative Institute for Research in Environmental Sciences (NOAA and University of Colorado Boulder)

CLIRAD: Climate and Radiation Branch, NASA Goddard Space Flight Center

CMIP5: Coupled Model Intercomparison Project Phase 5

CRIEPI: Central Research Institute of Electric Power Industry

CrIS: Cross-track Infrared Sounder

CRTM: Community Radiative Transfer Model

CTM: chemical transport model

DOE: Department of Energy

ECMWF: European Centre for Medium-Range Weather Forecasts

EMC: Environmental Modeling Center

EnKF: ensemble Kalman filter (a data assimilation scheme)

EOS: NASA's Earth Observing System

ERA-15: ECMWF 15-year reanalysis

ERA-20C: ECMWF 20th century reanalysis

ERA-40: ECMWF 40-year reanalysis

ERA5: a forthcoming reanalysis developed by ECMWF

ERA-Interim: ECMWF interim reanalysis

EUMETSAT: European Organisation for the Exploitation of Meteorological Satellites

ExUTLS: extra-tropical upper troposphere and lower stratosphere

FGGE: First GARP Global Experiment

GARP: Global Atmospheric Research Programme

GCOS: Global Climate Observing System

GEOS: Goddard Earth Observing System Model of the NASA

GFS: Global Forecast System of the NCEP

GLATOVS: Goddard Laboratory for Atmospheres TOVS (radiative transfer model)

GMAO: Global Modeling and Assimilation Office of NASA

GNNS-RO: Global Navigation Satellite System Radio Occultation (see also GPS-RO)

GOCART: Goddard Chemistry Aerosol Radiation and Transport

GOES: Geostationary Operational Environmental Satellite

GOME: Global Ozone Monitoring Experiment

GPS-RO: Global Positioning System Radio Occultation (see also GNSS-RO)

GRUAN: GCOS Reference Upper Air Network

GSICS: Global Space-based Inter-Calibration System

GSM: Global Spectral Model of the JMA

GTS: Global Telecommunication System

HALOE: Halogen Occultation Experiment

HCFC: hydrochlorofluorocarbon

HIRS: High-resolution Infrared Radiation Sounder

IASI: Infrared Atmospheric Sounding Interferometer

IAU: incremental analysis update

ICSU: International Council for Science

IFS: Integrated Forecast System of the ECMWF

IGY: International Geophysical Year

IOC: Intergovernmental Oceanographic Commission of UNESCO

JCDAS: JMA Climate Data Assimilation System

5 JMA: Japan Meteorological Agency

JRA-25: Japanese 25-year Reanalysis

JRA-55: Japanese 55-year Reanalysis

JRA-55AMIP: Japanese 55-year Reanalysis based on AMIP-type simulations

JRA-55C: Japanese 55-year Reanalysis assimilating Conventional observations only

10 LEO/GEO: Low Earth Orbit / Geostationary

McICA: Monte-Carlo independent column approximation

MERRA: Modern Era Retrospective-Analysis for Research

Meteosat: series of geostationary meteorological satellites operated by EUMETSAT

MIPAS: Michelson Interferometer for Passive Atmospheric Sounding

15 MISR: Multi-angle Imaging SpectroRadiometer

MLS: Microwave Limb Sounder

MODIS: Moderate resolution Imaging Spectroradiometer

MRF: Medium Range Forecast Version of the NCEP GFS

MRI-CCM1: Meteorological Research Institute (JMA) Chemistry Climate Model, version 1

20 MSU: Microwave Sounding Unit

MTSAT: Multi-functional Transport Satellite

NASA: National Aeronautics and Space Administration

NCAR: National Center for Atmospheric Research

NCEP: National Centers for Environmental Prediction of the NOAA

25 NMC: National Meteorological Center (now NCEP)

NOAA: National Oceanic and Atmospheric Administration

OMI: Ozone Monitoring Instrument

PAOBS: synthetic surface pressure data produced for the southern hemisphere by the Australian BOM

PIBAL: pilot balloon

30 QBO: quasi-biennial oscillation

R1: NCEP-NCAR Reanalysis 1

R2: NCEP-DOE Reanalysis 2

RAOBCORE: Radiosonde Observation Correction using Reanalyses

RH: relative humidity

RRTM: Rapid Radiative Transfer Model developed by Atmospheric and Environmental Research

RRTMG: RRTM for application to GCMs

RTTOV: Radiative Transfer for TOVS

SAO: semi-annual oscillation

SBUV: Solar Backscatter Ultraviolet Radiometer

SCIAMACHY: Scanning Imaging Absorption Spectrometer for Atmospheric Chartography

SHIP: surface meteorological observation report from ships

SOLARIS: Solar Influences for SPARC

SPARC: Stratosphere-troposphere Processes And their Role in Climate

S-RIP: SPARC Reanalysis Intercomparison Project

SSM/I or SSMI: Special Sensor Microwave Imager

SSMIS: Special Sensor Microwave Imager Sounder

SST: sea surface temperature

SSU: Stratospheric Sounding Unit

SYNOP: surface meteorological observation report from manned and automated weather stations

TCO: total column ozone

Terra: an EOS satellite

TIM: Total Irradiance Monitor

TIROS: Television Infrared Observation Satellite

TMI: Tropical Rainfall Measuring Mission (TRMM) Microwave Imager

TOA: top of atmosphere

TOMS: Total Ozone Mapping Spectrometer

TOVS: TIROS Operational Vertical Sounder

TSI: total solar irradiance

TTL: tropical tropopause layer

UARS: Upper Atmosphere Research Satellite

UNESCO: United Nations Educational, Scientific and Cultural Organization

USLM: upper stratosphere and lower mesosphere

UTLS: upper troposphere and lower stratosphere

VTPR: Vertical Temperature Profile Radiometer

WCRP: World Climate Research Programme

WMO: World Meteorological Organization

*Acknowledgements.* We acknowledge the scientific guidance and sponsorship of the World Climate Research Programme, coordinated in the framework of SPARC. We are grateful to Ted Shepherd and Greg Bodeker (past SPARC co-chairs) for strong encouragement and valuable advice during the proposal phase of the project during 2011–2012, and Joan Alexander (serving SPARC co-chair during 2013–2015) and Neil Harris (serving SPARC co-chair since 2014) for continued support and encouragement. The Information Initiative Center of Hokkaido University, Japan has hosted the S-RIP web server since 2014. We thank the reanalysis centres for providing their support and data products. The British Atmospheric Data Centre (BADC) of the UK Centre for Environmental Data Analysis (CEDA) has provided a virtual machine for data processing and a group workspace for data storage. We thank Yulia Zyulyaeva for contributions to S-RIP as an original member of the working group, as a chapter co-lead through October 2014, and as the designer of the S-RIP logo. We thank Diane Pendlebury for contributions to S-RIP as a chapter co-lead through August 2015. We thank Quentin Errera, the lead of the SPARC Data Assimilation working group, for co-organizing the 2014, 2015, and 2016 S-RIP workshops. Travel support for some participants of the 2013 planning meeting and the 2014, 2015, and 2016 workshops was provided by SPARC. We thank Peter Haynes, Gabriele Stiller, and William Lahoz for serving as the editors for the special issue "The SPARC Reanalysis Intercomparison Project (S-RIP)" in this journal. The materials contained in the tables and figures for the technical aspects of the reanalysis data sets have been compiled from a variety of sources, for which we acknowledge the contributions of Santha Akella, Michael Bosilovich, Dick Dee, John Derber, Ron Gelaro, Yu-Tai Hou, Robert Kistler, Daryl Kleist, Shinya Kobayashi, Shrinivas Moorthi, Eric Nielsen, Paul Poli, Bill Putman, Suranjana Saha, Jack Woollen, Fanglin Yang, and Valery Yudin. We thank Kazuyuki Miyazaki and Karen Rosenlof for valuable comments and suggestions on this manuscript. Support for the Twentieth Century Reanalysis Project dataset is provided by the U.S. Department of Energy, Office of Science Innovative and Novel Computational Impact on Theory and Experiment (DOE INCITE) program, and Office of Biological and Environmental Research (BER), and by the National Oceanic and Atmospheric Administration Climate Program Office. MF's contribution was financially supported in part by the Japanese Ministry of Education, Culture, Sports, Science and Technology (MEXT) through Grants-in-Aid for Scientific Research (26287117 and 16K05548). Work at the Jet Propulsion Laboratory, California Institute of Technology, was carried out under a contract with the National Aeronautics and Space Administration. EPG acknowledges support from the US NSF. VLH was supported by NSF CEDAR grant 1343056 and NASA LWS grant NNX14AH54G.

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

**Table 1:** List of global atmospheric reanalysis systems discussed in this work.

| Reanalysis system | Reference | Description |
| --- | --- | --- |
| ERA-40 | Uppala et al. (2005) | Centre: ECMWF<br>Coverage: September 1957 to August 2002 |
| ERA-Interim | Dee et al. (2011) | Centre: ECMWF<br>Coverage: January 1979 to present |
| ERA-20C[a] | Poli et al. (2016) | Centre: ECMWF<br>Coverage: January 1900 to December 2010 |
| JRA-25 / JCDAS (JRA-25) | Onogi et al. (2007) | Centre: JMA and CRIEPI<br>Coverage: January 1979 to January 2014 |
| JRA-55[b] | Kobayashi et al. (2015) | Centre: JMA<br>Coverage: January 1958 to present |
| MERRA | Rienecker et al. (2011) | Centre: NASA GMAO<br>Coverage: January 1979 to February 2016 |
| MERRA-2[c] | Bosilovich et al. (2015) | Centre: NASA GMAO<br>Coverage: January 1980 to present |
| NCEP-NCAR R1 (R1) | Kalnay et al. (1996); Kistler et al. (2001) | Centre: NOAA/NCEP and NCAR<br>Coverage: January 1948 to present |
| NCEP-DOE R2 (R2) | Kanamitsu et al. (2002) | Centre: NOAA/NCEP and the DOE AMIP-II project<br>Coverage: January 1979 to present |
| CFSR (CDAS-T382) | Saha et al. (2010) | Centre: NOAA/NCEP<br>Coverage: January 1979 to December 2010 |
| CFSv2 (CDAS-T574) | Saha et al. (2014) | Centre: NOAA/NCEP<br>Coverage: January 2011 to present |
| NOAA-CIRES 20CR v2[d] (20CR) | Compo et al. (2011) | Centre: NOAA and the University of Colorado CIRES<br>Coverage: November 1869 to December 2012 |

[a]A companion ensemble of AMIP simulations is also available: ERA-20CM; see Section 2 for details.

[b]Two ancillary products are also available: JRA-55C and JRA-55AMIP; see Section 2 for details.

5 [c]A companion AMIP simulation for MERRA-2 is in progress, but has not yet been completed as of this writing.

[d]A new version of 20CR covering 1851–2011 (20CR v2c) has been completed and made available in 2015.

**Table 2.** Basic specifications of the reanalysis forecast models. Approximate longitude grid spacing is reported in degrees for models with regular Gaussian grids (F*n*) and in kilometres for models with reduced Gaussian grids (N*n*). Wavenumber truncations for models with Gaussian grids are reported in parentheses.

| Reanalysis system | Model[a] | Horizontal grid spacing | Vertical levels | Top level |
|---|---|---|---|---|
| ERA-40 | IFS Cycle 23r4 (2001) | N80 ($T_L$159): ~125 km | 60 (hybrid $\sigma$–$p$) | 0.1 hPa |
| ERA-Interim | IFS Cycle 31r2 (2007) | N128 ($T_L$255): ~79 km | 60 (hybrid $\sigma$–$p$) | 0.1 hPa |
| ERA-20C | IFS Cycle 38r1 (2012) | N80 ($T_L$159): ~125 km | 91 (hybrid $\sigma$–$p$) | 0.01 hPa |
| JRA-25 | JMA GSM (2004) | F80 (T106): 1.125° | 40 (hybrid $\sigma$–$p$) | 0.4 hPa |
| JRA-55 | JMA GSM (2009) | N160 ($T_L$319): ~55 km | 60 (hybrid $\sigma$–$p$) | 0.1 hPa |
| MERRA | GEOS 5.0.2 (2008) | $^1/_2$° latitude × $^2/_3$° longitude | 72 (hybrid $\sigma$–$p$) | 0.01 hPa |
| MERRA-2 | GEOS 5.12.4 (2015) | 0.5° latitude × 0.625° longitude | 72 (hybrid $\sigma$–$p$) | 0.01 hPa |
| R1 | NCEP MRF (1995) | F47 (T62): 1.875° | 28 ($\sigma$) | 3 hPa |
| R2 | Modified MRF (1998) | F47 (T62): 1.875° | 28 ($\sigma$) | 3 hPa |
| CFSR (CDAS-T382) | NCEP CFS (2007) | F288 (T382): 0.3125° | 64 (hybrid $\sigma$–$p$) | ~0.266 hPa |
| CFSv2 (CDAS-T574) | NCEP CFS (2011) | F440 (T574): 0.2045° | 64 (hybrid $\sigma$–$p$) | ~0.266 hPa |
| 20CR | NCEP GFS (2008) | F47 (T62): 1.875° | 28 (hybrid $\sigma$–$p$) | ~2.511 hPa |

5  [a]Year in parentheses indicates the year for the version of the operational analysis system that was used for the reanalysis.

**Table 3:** Major physical parametrizations in the reanalysis forecast models. Radiation parametrizations are divided into shortwave (SW) and longwave (LW), cloud parametrizations are divided into convective (CU) and non-convective (LS), and gravity wave drag parametrizations are divided into orographic (ORO) and non-orographic (NON) components.

| Reanalysis | Radiation | Clouds | Gravity wave drag | Ozone model |
|---|---|---|---|---|
| ERA-40 | SW: Fouquart and Bonnel (1980) LW: Mlawer et al. (1997) | CU: Tiedtke (1989) LS: Tiedtke (1993) | ORO: Lott and Miller (1997) NON: none | Cariolle and Déqué (1986); Dethof and Hólm (2004) |
| ERA-Interim | SW: Fouquart and Bonnel (1980) LW: Mlawer et al. (1997) | CU: Tiedtke (1989) LS: Tiedtke (1993) | ORO: Lott and Miller (1997) NON: none | Cariolle and Déqué (1986); Dethof and Hólm (2004); Cariolle and Teyssèdre (2007) |
| ERA-20C | SW: Morcrette et al. (2008) LW: Morcrette et al. (2008) | CU: Tiedtke (1989) LS: Tiedtke (1993) | ORO: Lott and Miller (1997) NON: Scinocca (2003) | Cariolle and Déqué (1986); Dethof and Hólm (2004); Cariolle and Teyssèdre (2007) |
| JRA-25 | SW: Briegleb (1992) LW: Goody (1952) | CU: Arakawa and Schubert (1974) LS: Kawai and Inoue (2006) | ORO: Iwasaki et al. (1989a, 1989b) NON: none | Shibata et al. (2005) |
| JRA-55 | SW: Briegleb (1992); Freidenreich and Ramaswamy (1999) LW: Chou et al. (2001) | CU: Arakawa and Schubert (1974); Xie and Zhang (2000) LS: Kawai and Inoue (2006) | ORO: Iwasaki et al. (1989a, 1989b) NON: none | Shibata et al. (2005) |
| MERRA | SW: Chou and Suarez (1999) LW: Chou et al. (2001) | CU: Moorthi and Suarez (1992) LS: Bacmeister et al. (2006) | ORO: McFarlane (1987) NON: Garcia and Boville (1994) | Rienecker et al. (2008) |
| MERRA-2 | SW: Chou and Suarez (1999) LW: Chou et al. (2001) | CU: Moorthi and Suarez (1992) LS: Bacmeister et al. (2006) | ORO: McFarlane (1987) NON: Garcia and Boville (1994); Molod et al. (2015) | Rienecker et al. (2008) |
| R1 | SW: Lacis and Hansen (1974) LW: Fels and Schwarzkopf (1975); Schwarzkopf and Fels (1991) | CU: Arakawa and Schubert (1974); Tiedtke (1989) LS: grid-scale RH | ORO: Palmer et al. (1986); Pierrehumbert (1987); Helfand et al. (1987) NON: none | none |
| R2 | SW: Lacis and Hansen (1974) LW: Fels and Schwarzkopf (1975); Schwarzkopf and Fels (1991) | CU: Arakawa and Schubert (1974); Tiedtke (1989) LS: grid-scale RH | ORO: Palmer et al. (1986); Pierrehumbert (1987); Helfand et al. (1987) NON: none | none |
| CFSR | SW: Clough et al. (2005) LW: Clough et al. (2005) | CU: Tiedtke (1983); Moorthi et al (2001) LS: Xu and Randall (1996); Zhao and Carr (1997) | ORO: Kim and Arakawa (1995); Lott and Miller (1997) NON: none | McCormack et al. (2006) |
| CFSv2 | SW: Clough et al. (2005) LW: Clough et al. (2005) | CU: Tiedtke (1983); Moorthi et al (2001) LS: Xu and Randall (1996); Zhao and Carr (1997) | ORO: Kim and Arakawa (1995); Lott and Miller (1997) NON: Chun and Baik (1998) | McCormack et al. (2006) |
| 20CR | SW: Clough et al. (2005) LW: Clough et al. (2005) | CU: Tiedtke (1983); Moorthi et al (2001) LS: Xu and Randall (1996); Zhao and Carr (1997) | ORO: Kim and Arakawa (1995); Lott and Miller (1997) NON: none | McCormack et al. (2006) |

**Table 4:** Sources of SST and sea ice lower boundary conditions used in reanalyses.

| Reanalysis system | Dataset | Grid | Time | Reference |
|---|---|---|---|---|
| ERA-40 | HadISST1 (09/1957–11/1981) | 1° | monthly | Rayner et al. (2003) |
| | NCEP 2DVar (12/1981–06/2001) | 1° | weekly | Reynolds et al. (2002) |
| | NOAA OISSTv2 (07/2001–08/2002) | 0.25° | daily | Reynolds et al. (2007) |
| ERA-Interim | HadISST1 (09/1957–11/1981) | 1° | monthly | Rayner et al. (2003) |
| | NCEP 2DVar (12/1981–06/2001) | 1° | weekly | Reynolds et al. (2002) |
| | NOAA OISSTv2 (07/2001–12/2001) | 0.25° | daily | Reynolds et al. (2007) |
| | NCEP RTG (01/2002–01/2009) | 0.083° | daily | Gemmill et al. (2007) |
| | OSTIA (02/2009–present) | 0.05° | daily | Donlon et al. (2012) |
| ERA-20C | HadISSTv2.1.0.0 | 0.25° | daily | Titchner and Rayner (2014) |
| JRA-25 / JCDAS | COBE | 1° | daily | Ishii et al. (2005) |
| | NH sea ice analysis | | | Walsh and Chapman (2001) |
| | SH sea ice analysis | | | Matsumoto et al. (2006) |
| JRA-55[a] | COBE | 1° | daily | Ishii et al. (2005) |
| | NH sea ice analysis | | | Walsh and Chapman (2001) |
| | SH sea ice analysis (after 10/1978) | | | Matsumoto et al. (2006) |
| MERRA | Hadley Centre (01/1979–12/1981) | 1° | monthly | n/a (personal communication) |
| | NOAA OISSTv2 (01/1982–present) | 1° | weekly | Reynolds et al. (2002) |
| MERRA-2 | AMIP-II (01/1980–12/1981) | 1° | monthly | Taylor et al. (2000) |
| | NOAA OISSTv2 (01/1982–03/2006) | 0.25° | daily | Reynolds et al. (2007) |
| | OSTIA (04/2006–present) | 0.05° | daily | Donlon et al. (2012) |
| NCEP-NCAR R1 | SSTs: | | | |
| | Met Office GISST (01/1948–10/1981) | 1° | monthly | Parker et al. (1995) |
| | NOAA OISSTv1 (11/1981–12/1994) | 1° | weekly | Reynolds and Smith (1994) |
| | NOAA OISSTv1 (01/1995–present) | 1° | daily | Reynolds and Smith (1994) |
| | Sea ice: | | | |
| | Navy/NOAA JIC (01/1948–10/1978) | varies | varies | Kniskern (1991) |
| | SMMR and SSM/I (11/1978–present) | 25 km | monthly | Grumbine (1996) |
| NCEP-DOE R2 | AMIP-II (01/1979–08/15/1999) | 1° | monthly | Taylor et al. (2000) |
| | NOAA OISSTv1 (08/16/1999–12/1999) | 1° | monthly | Reynolds and Smith (1994) |
| | NOAA OISSTv1 (01/2000–present) | 1° | daily | Reynolds and Smith (1994) |
| CFSR / CFSv2[a] | HadISST1.1 (01/1979–10/1981) | 1° | monthly | Rayner et al. (2003) |
| | NOAA OISSTv2 (11/1981–present) | 0.25° | daily | Reynolds et al. (2007) |
| NOAA-CIRES 20CR v2[b] | HadISST1.1 | 1° | monthly | Rayner et al. (2003) |

[a] A climatology is used for sea ice in the Southern Hemisphere in JRA-55 prior to October 1978 (Kobayashi et al., 2015).

[b] CFSR and CFSv2 produce SST analyses, but relax these internal products to the external SST analyses listed here (Saha et al., 2010).

5   [c] Sea ice concentrations were mis-specified in coastal regions during the production of 20CR (Compo et al., 2011).

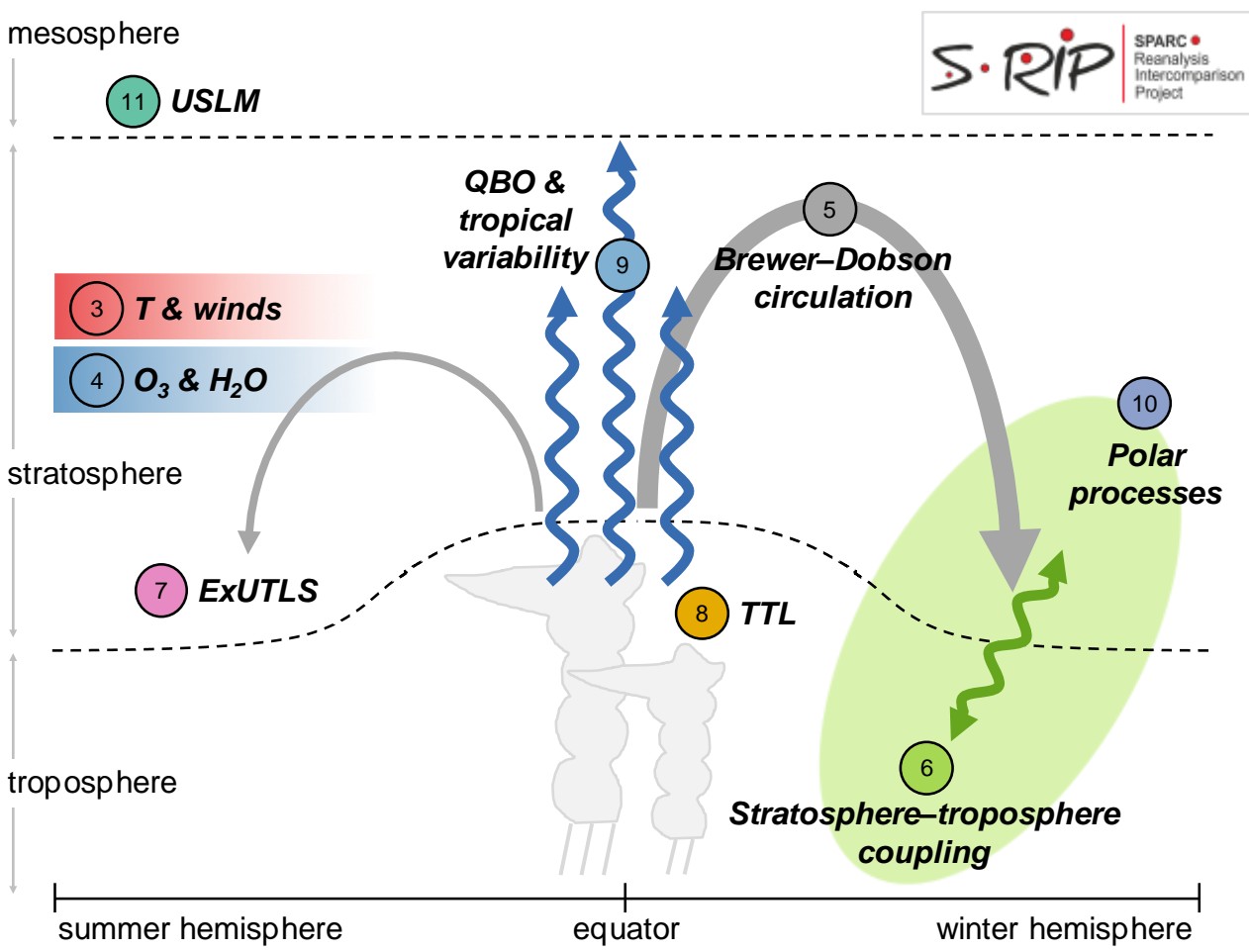

**Figure 1:** Schematic illustration of the atmosphere showing the processes and regions that will be covered by chapters in the planned full S-RIP report. Domains approximate the main focus areas of each chapter and should not be interpreted as strict boundaries. Chapters 3 and 4 cover the entire domain.

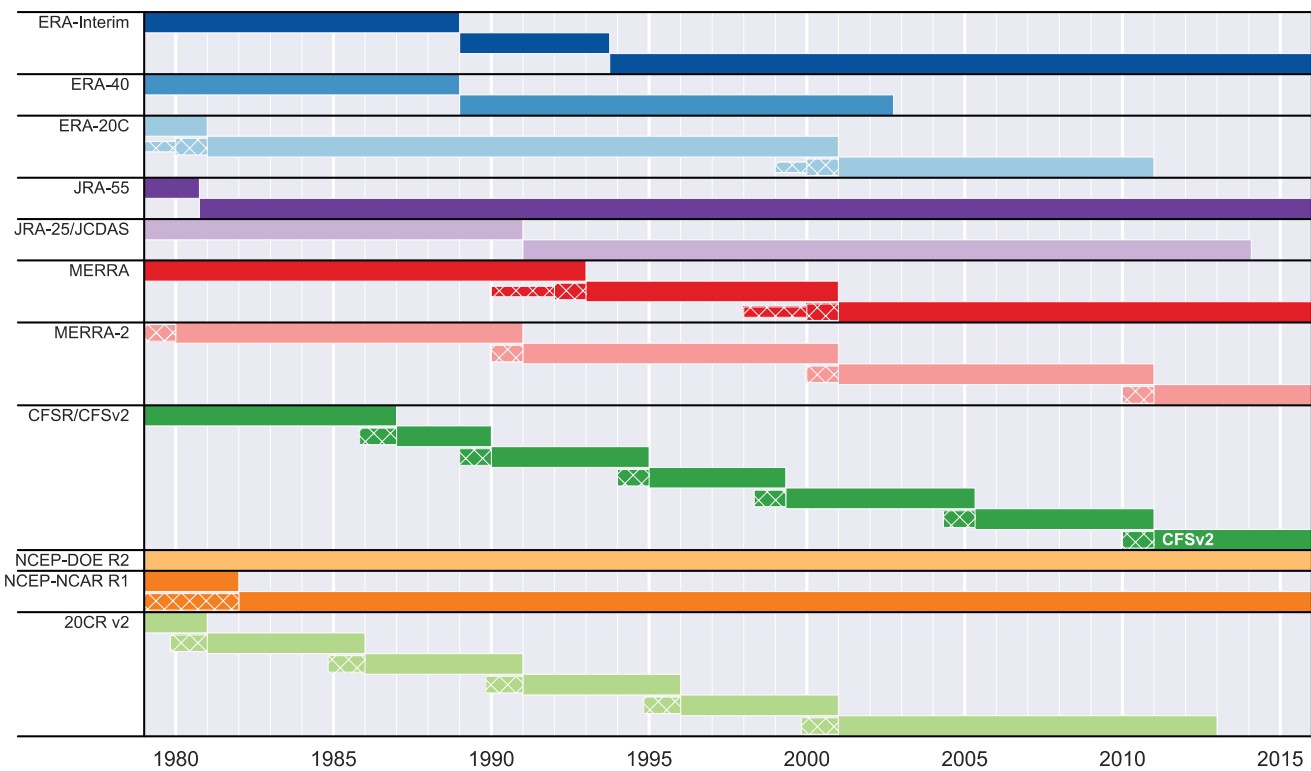

**Figure 2.** Summary of the execution streams of the reanalysis systems from January 1979 through December 2015. The narrowest cross-hatched sections indicate known spin-up periods, while the medium-narrow cross-hatched sections indicate overlap periods.

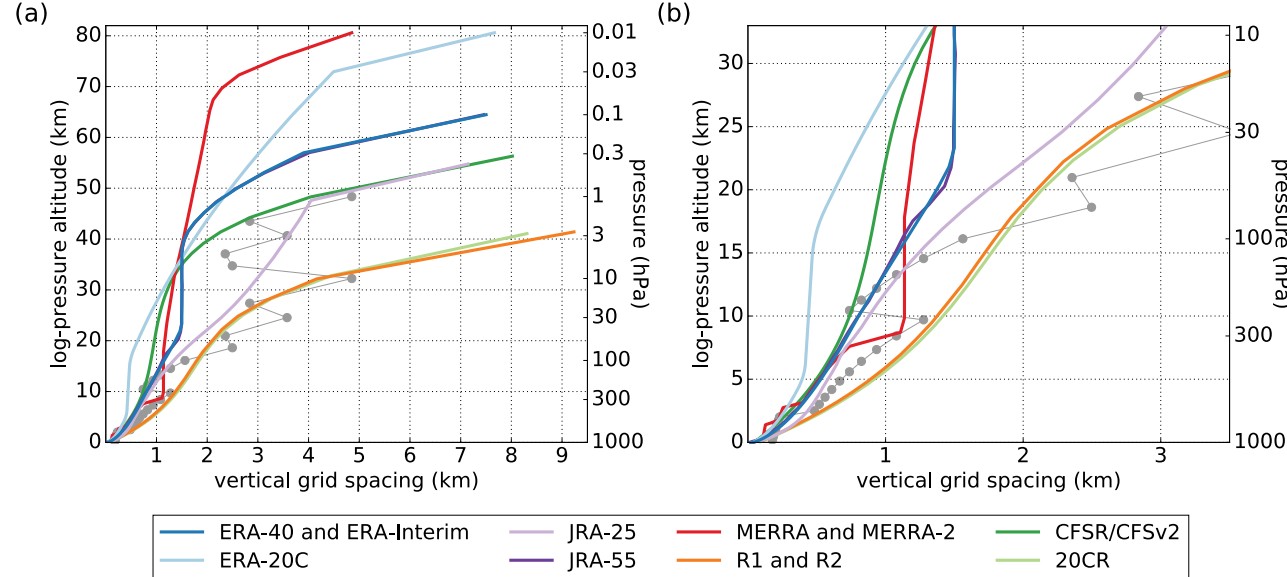

**Figure 3.** Approximate vertical resolutions of the reanalysis forecast models for (a) the full vertical range of the reanalyses and (b) the surface to 33 km (~10 hPa). Altitude and vertical grid spacing are estimated using log-pressure altitudes ($z^* = H \ln[p_0/p]$), where the surface pressure $p_0$ is set to 1000 hPa and the scale height $H$ is set to 7 km. The grid spacing indicating the separation of two levels is plotted at the altitude of the upper of the two levels, so that the highest altitude shown in (a) indicates the lid location. Some reanalyses use identical vertical resolutions; these systems are listed together in the legend. Other reanalyses have very similar vertical resolutions when compared with other systems, including JRA-55 (similar but not identical to ERA-40 and ERA-Interim) and 20CR (similar but not identical to R1 and R2). Approximate vertical spacing associated with the isobaric levels on which ERA-40 and ERA-Interim reanalysis products are provided (grey discs) is shown in both panels for context.

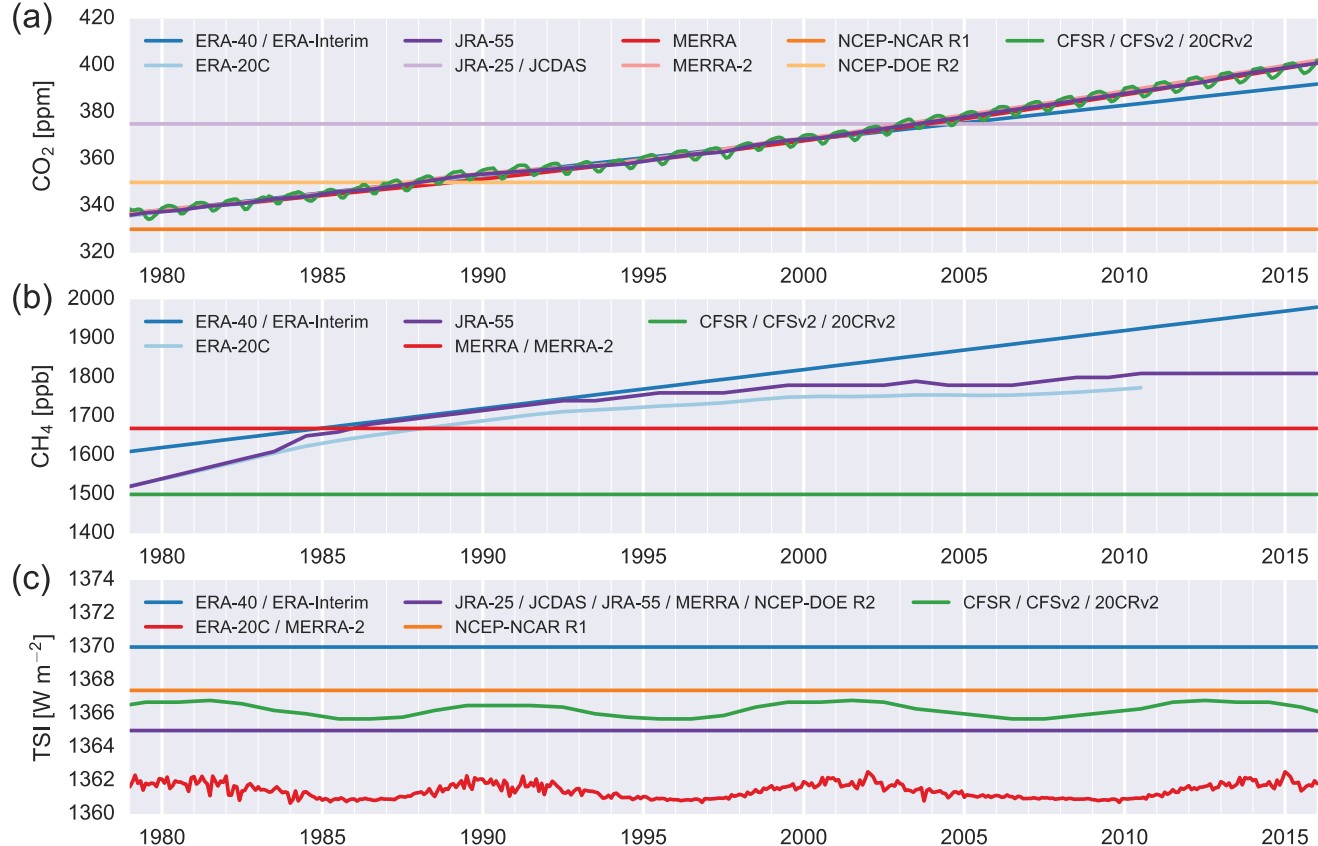

**Figure 4.** Time series of boundary conditions for (a) $CO_2$, (b) $CH_4$, and (c) TSI used by the reanalysis systems from 1979 through 2015. The $CH_4$ climatology used in MERRA and MERRA-2 varies in both latitude and height; here a "tropospheric mean" value is calculated as a mass- and area-weighted integral between 1000 and 288 hPa to facilitate comparison with the "well-mixed" values used by most other systems. ERA-20C also applies rescalings of annual mean values of both $CO_2$ and $CH_4$ that vary by latitude, height, and month; here the base (global annual mean) values are shown. Time series of TSI neglect seasonal variations due to the ellipiticity of the Earth's orbit, as these variations are applied similarly (but not identically) across reanalysis systems. See text for further details.

**a** 3D-Var (increments calculated and applied at analysis times)

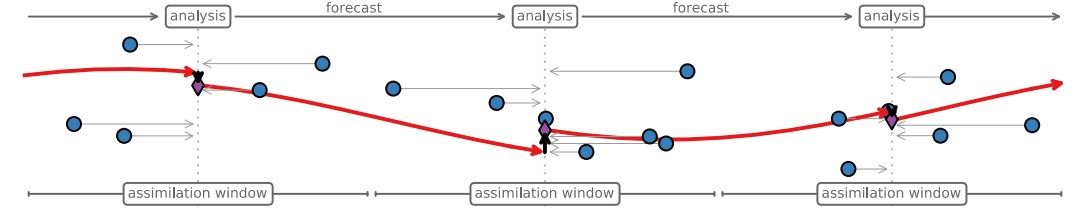

**b** 3D-FGAT (increments estimated at observation times but applied at analysis times)

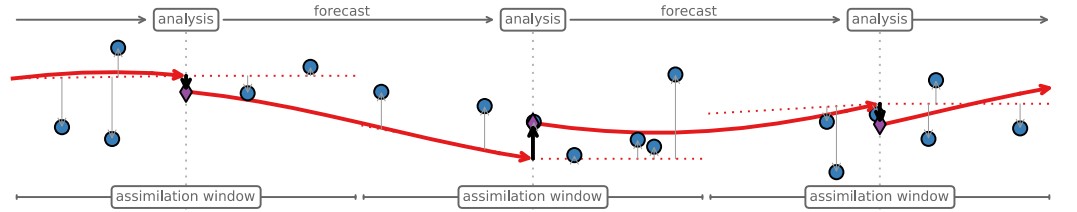

**c** 4D-Var (iteratively estimate increments for full window and adjust initial state)

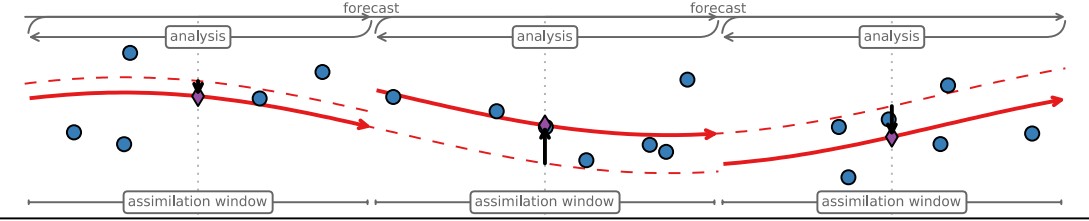

**d** EnKF (increment applied as a Bayesian update to the posterior forecast ensemble)

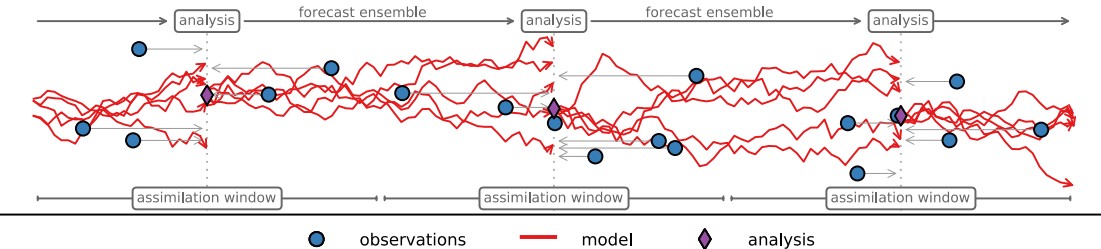

**Figure 5.** Simplified schematic representations of four data assimilation strategies used by current reanalyses: (a) 3D-Var, (b) 3D-FGAT, (c) incremental 4D-Var, and (d) EnKF. Blue circles represent observations, red lines represent model trajectories, and purple diamonds indicate analyses. The dotted red lines in (b) represent linearly interpolated/extrapolate forecast values used to estimate increments at observation times. The dashed red lines in (c) represent the initial forecast, prior to iterative adjustments. These illustrations are conceptual, and do not accurately reflect the much more complex strategies used by reanalysis systems.

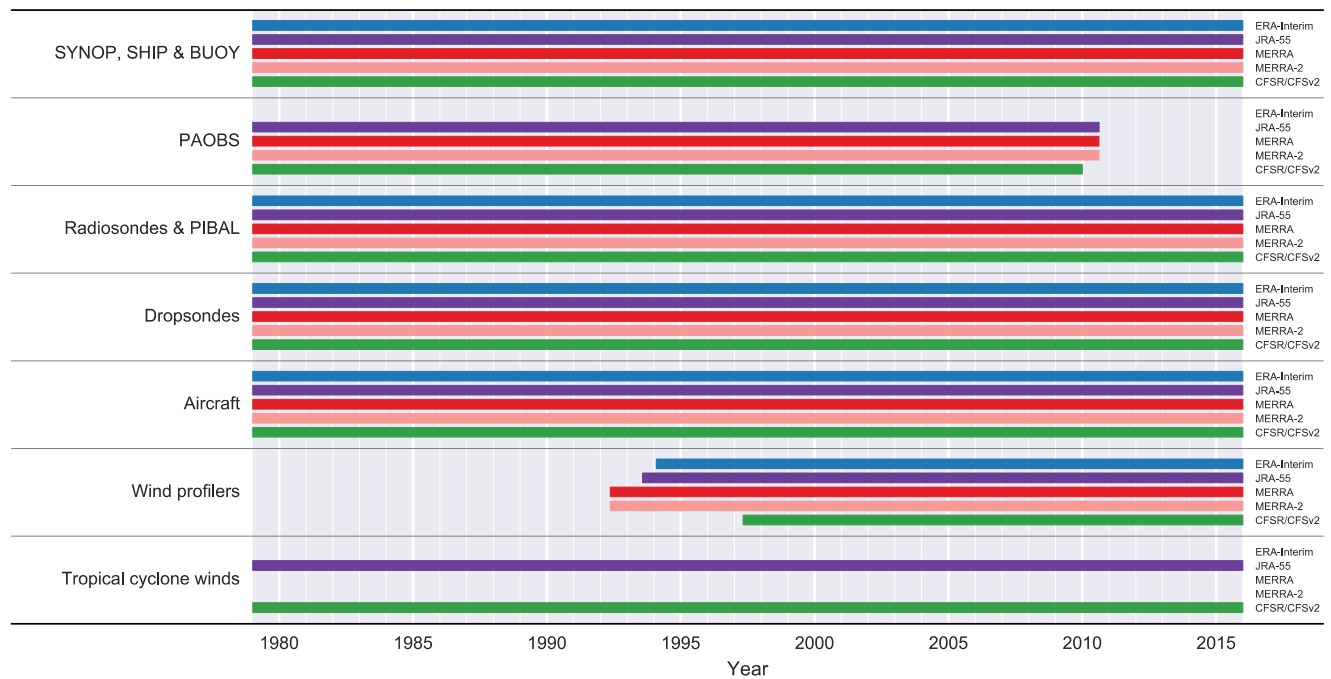

5  **Figure 6.** Timelines of conventional observations assimilated by the ERA-Interim (blue), JRA-55 (purple), MERRA (dark red), MERRA-2 (light red), and CFSR (green) reanalysis systems. See Appendix A for acronym definitions.

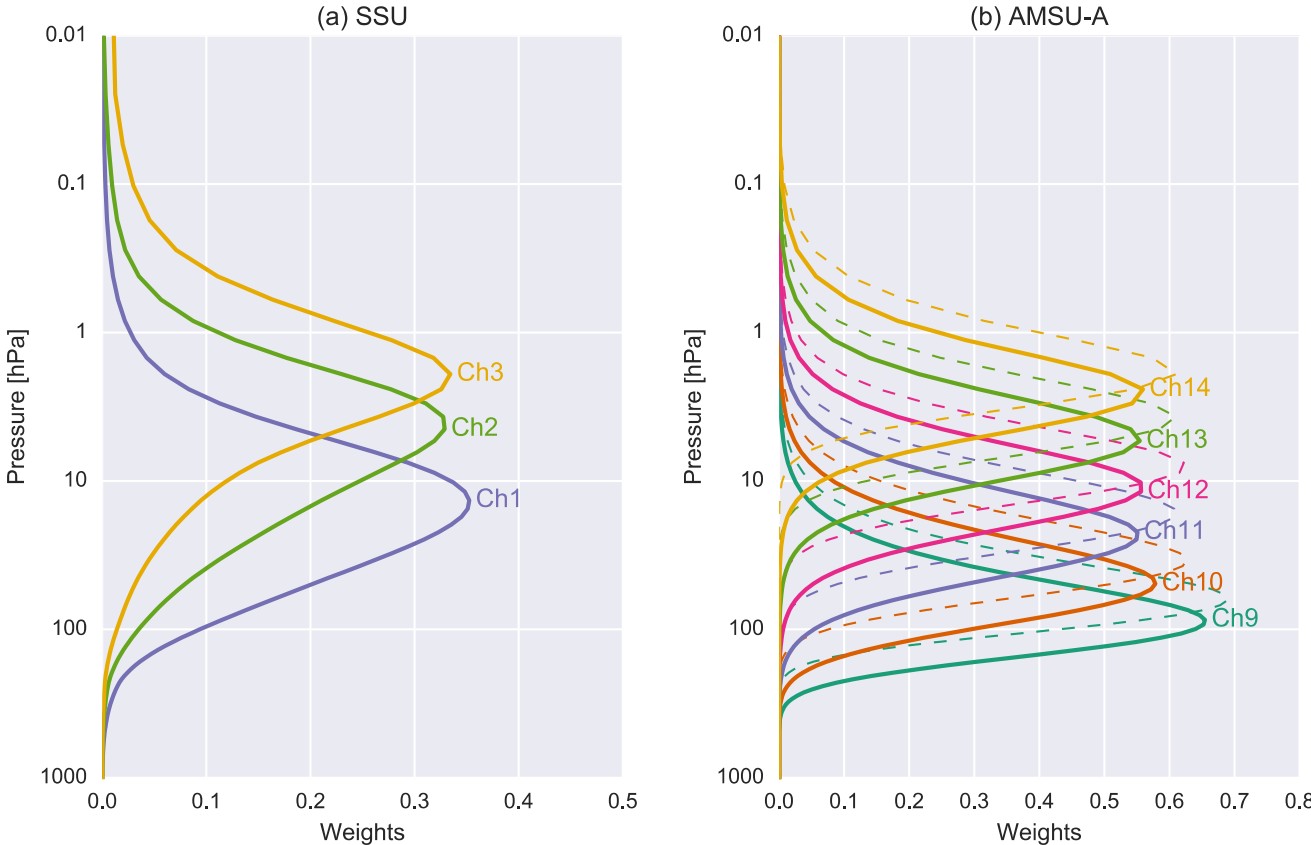

**Figure 7.** Vertical weighting functions of radiance measurements for (a) the TOVS-suite Stratospheric Sounding Unit (SSU) instrument (1979–2005) channel 1 (centred at ~15 hPa), channel 2 (~5 hPa), and channel 3 (~1.5 hPa), and (b) the ATOVS-suite Advanced Microwave Sounding Unit A (AMSU-A) instrument (1998–present) temperature channels 9–14 at near nadir (1.67°, solid lines) and limb (48.33°, dashed lines) scan positions. SSU channels 1 through 3 may also be referred to as TOVS channels 25 through 27.

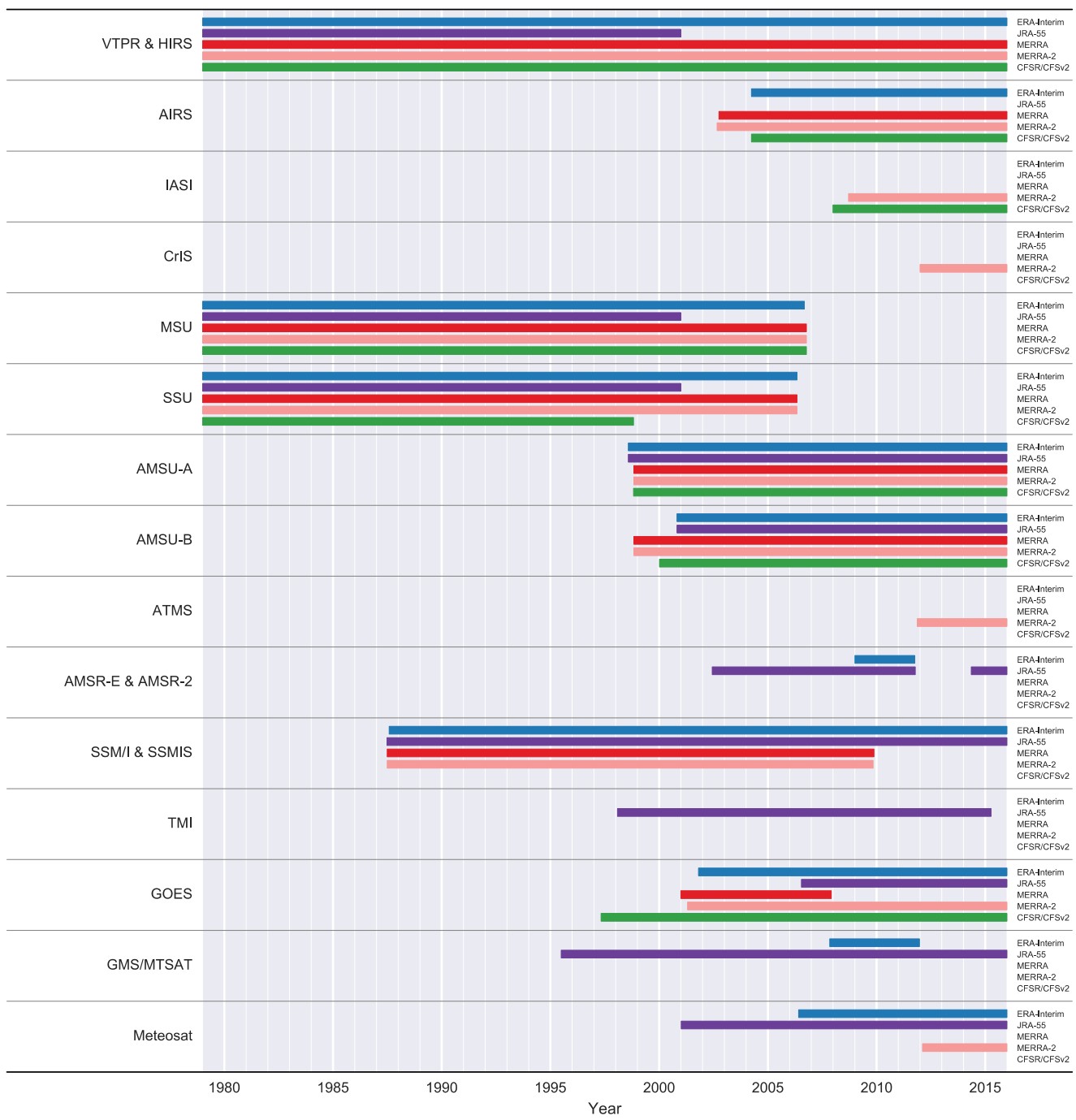

**Figure 8.** As in Fig. 6, but for satellite radiance observations assimilated by the reanalysis systems. See Appendix A for acronym definitions.

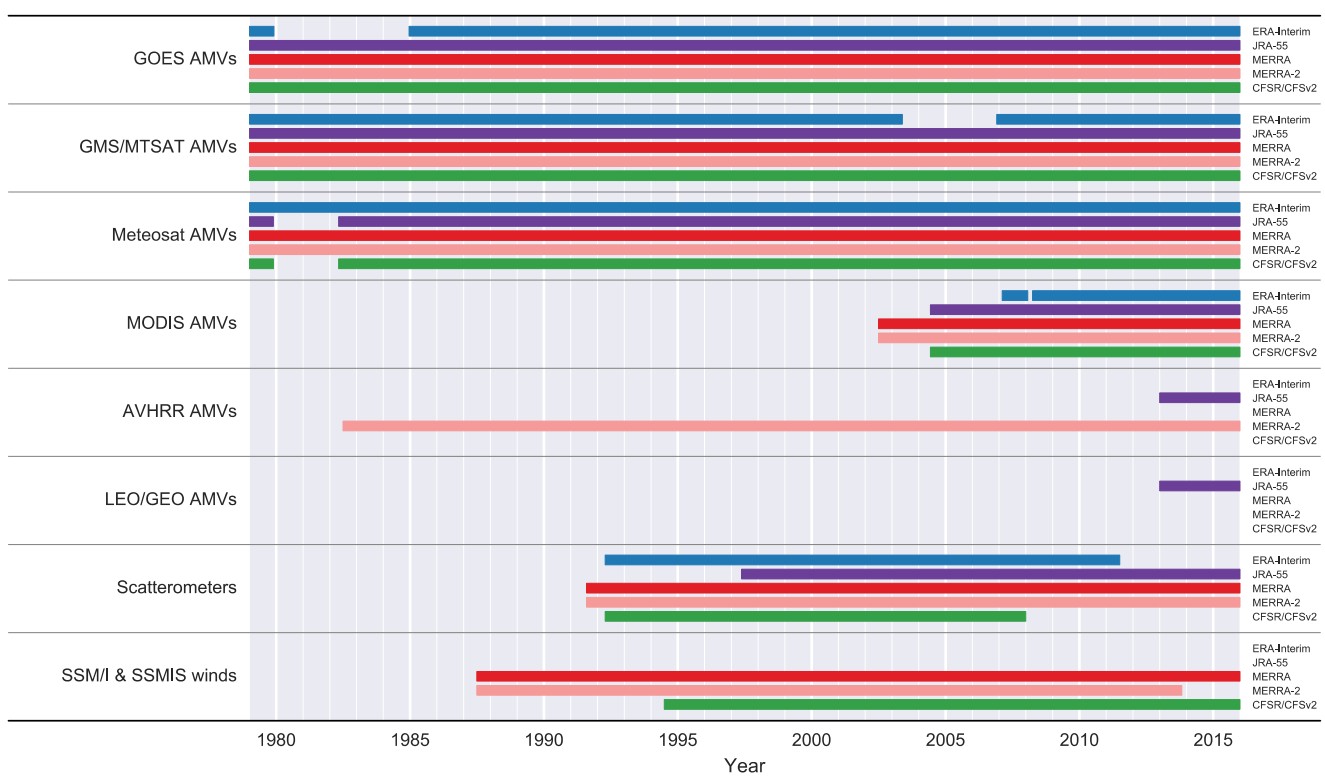

**Figure 9.** As in Fig. 6, but for AMVs and ocean surface wind products derived from satellites and assimilated by the reanalysis systems. See Appendix A for acronym definitions.

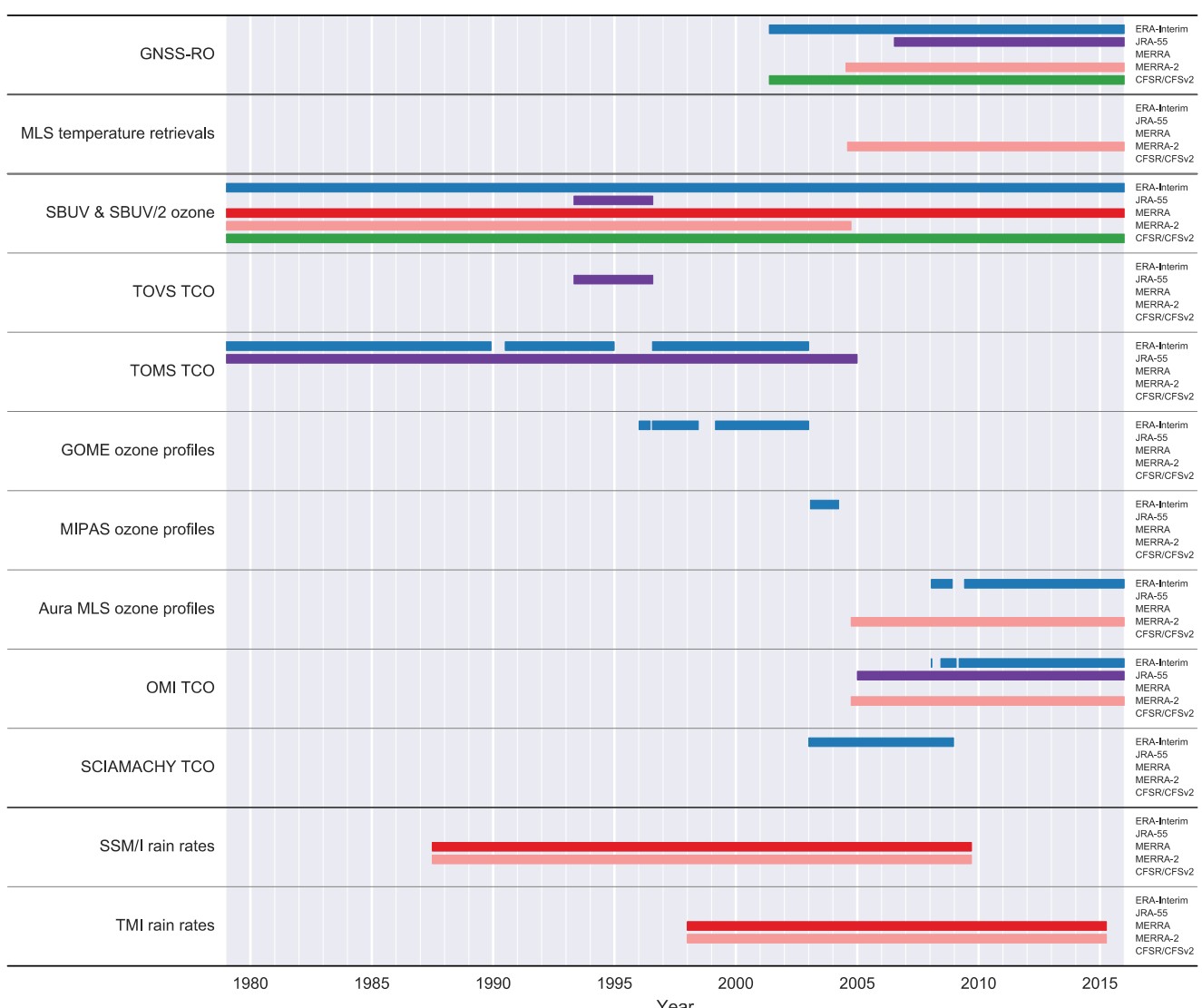

**Figure 10.** As in Fig. 6, but for other types of satellite observations assimilated by the reanalysis systems, including GNSS-RO (or GPS-RO), satellite temperature and ozone retrievals, and rain rates derived from microwave imagers. Aura MLS temperature retrievals are assimilated by MERRA-2 only in the upper stratosphere at pressures less than 5 hPa. SBUV and SBUV/2 are used by reanalyses to supply ozone profile information, TCO, or both. JRA-55 assimilates only TCO, ERA-Interim assimilates only profiles, and CFSR, MERRA, and MERRA-2 assimilate both TCO and profiles. See Appendix A for acronym definitions.