# Peer review of "Introduction to the SPARC Reanalysis Intercomparison Project (S-RIP) and overview of the reanalysis systems"

_Atmospheric Chemistry and Physics, 2016_

## Referee Comment (RC1) · Anonymous Referee #1 · 18 Aug 2016

Review of " Introduction to the SPARC Reanalysis Intercomparison Project (S-RIP) and overview of the reanalysis systems" by Fujiwara et al

This paper is a summary of reanalysis systems for a stratospheric inter comparison project. It is a useful summary, but could be much more focused on the implications of the qualitative comparison of reanalyses for the stratosphere. The manuscript hints at this, but there is too much exposition, and not enough high level analysis. This would should be suitable for publication with revisions.

Overall I learned a great deal about reanalysis systems. I think the work need not focus on SRIP programatic functions at the beginning and end (the table of report chapters is not relevant). This should be less a summary and preview of an SRIP report than a

stand alone qualitative summary of reanalyses: with implications for the stratosphere.

I would like to see some of the very nice comments sprinkled throughout the text about how the reanalysis system structure impacts the stratosphere put into the conclusions. For example: assimilation methods may introduce variance which affect transport and may induce spurious gravity wave momentum transport. These are mentioned, but this and other features of the systems could be better highlighted at the end.

As mentioned, table 1 could be deleted. Figure 1 is probably not discussed enough to be necessary. Also, Figure 6 I think is only mentioned once in passing, and may not be necessary (it is sufficient to state weighting functions are broad).

Specific comments

Along the lines above, the title might be better as "Overview of the reanalysis systems in the stratosphere for the SPARC Reanalysis Intercomparison Project (S-RIP)"

P2, L25: I think it would be wise to spell out the Acronyms where they first appear, as is typical custom. That is an editorial decision for ACP.

P3, L7: A list of 13 papers as examples is not really necessary. Probably just citing examples in Fujiwara et al 2012b is appropriate, unless you want to specifically describe any examples in a sentence.

P3, L22 through P4, L14: Describing a report is not helpful or necessary. These paragraphs, table 1 and figure 1 could be deleted.

P5, L6: Since the NOAA/NCEP systems are the oldest, maybe section 2.4 should be placed first in section 2.

P15, L5: here is a good summary about what impact assimilation and reanalysis system structure may have on results: suggest this result and other similar ones be part of the conclusions.

P17, L6: Is this the only mention of Figure 6? Suggest deleting it. All the figure shows

is the 'deep vertical weighting functions' you mention.

P18, L25: What is homogenized data?

P19, L22: State why GPS-RO is unbiased in a sentence or two.

P21, L33: What is the impact of the simplified H2O treatment on the stratospheric analysis?

P22, L10-12: How are each of these affected by the reanalysis systems themselves? This would be a good place to spend a few paragraphs in summary

P22, L15-31: I don't think project plans for SRIP belong in ACP.

---

## Referee Comment (RC2) · Anonymous Referee #2 · 29 Aug 2016

This is an overview document for the volume on the intercomparison of stratospheric
reanalyses – the Stratosphere–troposphere Processes And their Role in Climate
(SPARC) Reanalysis Intercomparison Project (S-RIP). The manuscript is very well writ-
ten, and achieves its primary goal of a useful description that can stand as reference
document for the papers in the volume. In fact, it leaves me looking forward to seeing
some results.

I hope the authors consider the following comments to complement the present
manuscript.

1) There have been previous intercomparisons of reanalyses, including some discus-
sion of the stratosphere. A paragraph mentioning some of these intercomparisons and
lessons learned would strengthen the context of the current paper.

2) In Section 3, Page 8, it is noted that a key difference between the assimilation systems is the height of the top level of the model and vertical resolution. What is not mentioned in the paper is the treatment of the upper layers of the model. I postulate that the scientific foundation of many of the differences that are revealed will be related to the upper boundary and its treatment. Therefore, it would be useful to tabulate whether or not there is a sponge layer, the depth of the sponge layer and the dissipative methods that are used in the sponge layer – Rayleigh friction, enhanced horizontal diffusion, etc. This information will be especially important in the discussion of the Brewer-Dobson circulation, and diagnostics such as the age of air.

3) Since one of the applications of stratospheric reanalyses is to understand tracer transport, it would be useful to have a table of how the models, in general, treat diffusion – second order, fourth order, Rayleigh friction, etc.

4) In both the text, Page 8, and the tables, the spatial size in km should be given for all of the models. Presently it is in degrees for some models and km for others. Consistent presentation of technical information should be checked throughout.

5) Coming from a numerical background, the equating of "resolution" with "grid cell size" always bothers me. Generally, I live with my idiosyncrasy. However, the statement in Section 3, Page 8, that the "effective horizontal resolution is ..." is one I cannot let past. Since, effective resolution of weather and climate models is a topic of some controversy, with several lines of research, this is a term than should be used correctly. This is grid size, not the effective resolution, which will be 6 to 10 times the grid size.

6) I would like to see tabulated information on the treatment of $CO_2$ and $CH_4$ (Page 9) and aerosols (Page 10) in the radiative schemes. Since these reanalysis cover a span of time when climate is, definitively, not stationary, the treatment of the greenhouse gases seems fundamental. From the text, the treatment varies greatly from model to model, and I can't differentiate one treatment from another in the current text. As

with the treatment of the upper layers of the model, I would expect the treatment of greenhouse gases and aerosols to have significant impact upon the science-based interpretation of the systems. Hence a quick summary table would be useful.

7) Similar to previous comment, I would like to see more description, table perhaps, of the sea-ice treatment in the models (Page 10). The changes in Arctic are large. There are trends in area of polar isotherms in the upper troposphere and lower stratosphere; more details of the treatment of the Arctic boundary conditions are needed.

8) In the sentences, page 13, the use of the word "attempt" to describe the 3-D – Var and 4-D – Var assimilation systems is peculiar. I assume that they do, in fact, do the optimization calculation.

9) Page 15. Like sea-ice, I could imagine a tabulation of how water vapor (Page 21) is treated in the upper troposphere and stratosphere would be useful for science-based interpretation.

10) Just by chance, I just saw that Jim Pfaendtner's name is misspelled in the Helfand reference (Page 31). It is spelled as "Pfaentner," and it should be "Pfaendtner."

---

## Author Response (AR1)

**Response to Anonymous Referee #1**

**I would like to see some of the very nice comments sprinkled throughout the text about how the reanalysis system structure impacts the stratosphere put into the conclusions.**

We have expanded this part of the discussion, and have also included a few additional points along these lines. See p.9, l.11–16; p.12, l.20–24; p.20, l.27 through p.21, l.2

**The title might be better as "Overview of the reanalysis systems in the stratosphere for the SPARC Reanalysis Intercomparison Project (S-RIP)"**

We have discussed this internally and have chosen to keep the original title because this paper also serves as the introductory paper for the S-RIP special issue and, as such, should briefly outline the mission and structure of the S-RIP project. We have also discussed the option of separating the paper into two parts, a short paper introducing S-RIP (and the special issue) and a longer paper reviewing the reanalysis systems (which would then take this or a similar title). This approach has its appeals, but would involve technical and logistical issues that we prefer to avoid unless the referees and the editor in charge have strong preferences for it.

**Table 1 could be deleted.**

Table 1 has been deleted.

**Figure 1 is probably not discussed enough to be necessary.**

Figure 1 is retained, to support the component of the paper that is intended to introduce the S-RIP special issue.

**P2, L25: I think it would be wise to spell out the Acronyms where they first appear, as is typical custom. That is an editorial decision for ACP.**

After discussion with the editor in charge, we have updated the manuscript to introduce more of the acronyms within the text. The modified approach is outlined at the end of the first paragraph of section 1 (p.2, l.28–30): "A key for all acronyms used in this paper is provided in Appendix A. Acronyms representing the names of institutes, models, satellites, and other entities are in most cases only provided in the appendix; all other acronyms are both introduced in the text and included in the appendix." The rationale is that many of these entities are already well known by 'brand name' acronyms. Given the length of the paper, we feel that readability is better served by brevity in these cases.

**P3, L7: A list of 13 papers as examples is not really necessary. Probably just citing examples in Fujiwara et al 2012b is appropriate, unless you want to specifically describe any examples in a sentence.**

We have replaced this with "a list of recent examples has been provided by Fujiwara et al., 2012; see also the contents of this special issue" (p.3, l.15–16).

**P3, L22 through P4, L14: Describing a report is not helpful or necessary. These paragraphs, table 1 and figure 1 could be deleted.**

The discussion here has been modified somewhat to remove unnecessary terminology (e.g., "basic" vs "advanced" chapters) and make the presentation more general. We have also added references to certain chapters of the interim report where more information is

provided (p.12, l.8–9; p.25, l.20–22; p.26, l.21–22).

**P5, L6: Since the NOAA/NCEP systems are the oldest, maybe section 2.4 should be placed first in section 2.**
> To account for the staggered release dates of the reanalyses, we have retained the original order: alphabetical by reanalysis centre, then chronological within each centre.

**P15, L5: here is a good summary about what impact assimilation and reanalysis system structure may have on results: suggest this result and other similar ones be part of the conclusions.**
> The potential for assimilation increments to generate spurious wave activity / instabilities is now repeated in the conclusions (p.26, l.4–8).

**P17, L6: Is this the only mention of Figure 6? Suggest deleting it. All the figure shows is the 'deep vertical weighting functions' you mention.**
> Figure 6 (now 7) is referenced multiple times in the revised text (p.19, l.33; p.20, l.29–31; p.26, l.15–18). This figure is particularly useful for illustrating the issues involved in the TOVS–ATOVS transition, which we have emphasized in the revised text as an example of the impacts that changes in the observing system can have on reanalysis products. This is a well-known issue amongst experts in the field, but not necessarily amongst the general climate community for whom this paper is intended.

**P18, L25: What is homogenized data?**
> We have added brief explanations to the text at p.18, l.16–17 ("in which observations from different launch sites and instrument suites are post-processed to remove biases, drifts, and jumps in the data record") and p.26, l.22–23 ("in which observations collected by different satellites are cross-calibrated to reduce biases and eliminate discontinuities in the data record"). The introduction to homogenized satellite radiances data on p.20 also mentions that homogenization in this context refers to "post-launch inter-satellite calibration".

**P19, L22: State why GPS-RO is unbiased in a sentence or two.**
> A brief explanation has been added on p.22, l.25–27: "GNSS-RO occultations are based on radio waves that are calibrated against on-board atomic clocks, and are therefore exceptionally stable both in time and across satellite platforms (Poli et al., 2010). The resulting retrievals have small random errors (equivalent to ~1 K) and very small systematic errors (less than ~0.2 K)."

**P21, L33: What is the impact of the simplified H2O treatment on the stratospheric analysis?**
> This is an open question, which we hope to address during S-RIP. In the meantime, we have added a paragraph at the end of section 6.2: "In general, reanalyses do not provide physically meaningful estimates of water vapour above the tropopause, although it should be noted that observational datasets used for comparison to the models have their own rather large biases in this region (Hegglin et al., 2013). Given the importance of water vapour in the upper troposphere and lower stratosphere (UTLS) for radiative forcing (e.g., Forster and Shine, 2002; Randel et al., 2007; Gettelmann et al., 2011; Riese et al., 2012), large biases in the representation of the water vapour gradients across the tropopause and in the lower stratosphere may lead to non-negligible radiative and dynamical impacts

in reanalysis systems. The magnitude of these impacts in the different reanalyses is not yet quantified, but is under investigation within S-RIP. Regardless, we emphasize in no uncertain terms that reanalysis humidity products in the upper troposphere and stratosphere should be used only with extreme caution."

**P22, L10-12: How are each of these affected by the reanalysis systems themselves? This would be a good place to spend a few paragraphs in summary**

We have described several examples that relate specifically to how data assimilation techniques, bias correction procedures, and changes in assimilated data may impact reanalysis products (see summary from p.26, l.1 through p.27, l.10). The core purpose of the S-RIP activity is to evaluate and document how each of these features are affected by the reanalysis systems. Many of these evaluations are currently in progress, but are not yet documented. We therefore defer a more systematic summary of how these features and regions of the middle and upper atmosphere are affected by the reanalysis systems to synthesis papers that will be written and submitted once the S-RIP report is completed.

**P22, L15-31: I don't think project plans for SRIP belong in ACP**

See response to title suggestion above.

**Response to Anonymous Referee #2**

**There have been previous intercomparisons of reanalyses, including some discussion of the stratosphere. A paragraph mentioning some of these intercomparisons and lessons learned would strengthen the context of the current paper.**

> In terms of comprehensive intercomparisons, the SPARC Intercomparison of Middle Atmosphere Climatologies is more than 10 years old, and includes only two of the reanalysis systems that we consider (NCEP-NCAR and ERA-40). We have added a sentence referencing that intercomparison (p.3, l.10–13). Quite a few subsequent studies have looked at certain aspects of reanalysis performance in the upper troposphere and stratosphere, including the 12 papers referenced in the original manuscript (removed at referee #1's suggestion) and several others published between 2004 and 2012. It is difficult to summarize these in a concise way, beyond the current note that "different results may be obtained for the same diagnostic" (p.3, l.14). In this revision, we have instead chosen to prioritize other material that is not as well documented (treatment of model upper layers, horizontal and vertical diffusion, boundary conditions, examples of how the structure of a reanalysis system propagates into the products of that system). Inclusion of a sufficiently comprehensive summary of lessons learned would likely require a more radical reorganization (such as splitting the paper into two; see response to referee #1).

**In Section 3, Page 8, it is noted that a key difference between the assimilation systems is the height of the top level of the model and vertical resolution. What is not mentioned in the paper is the treatment of the upper layers of the model. I postulate that the scientific foundation of many of the differences that are revealed will be related to the upper boundary and its treatment. Therefore, it would be useful to tabulate whether or not there is a sponge layer, the depth of the sponge layer and the dissipative methods that are used in the sponge layer – Rayleigh friction, enhanced horizontal diffusion, etc.**

> A description of the treatment of the upper layers in each model and a brief discussion of associated issues has been added (p.9, l.8–31).

**Since one of the applications of stratospheric reanalyses is to understand tracer transport, it would be useful to have a table of how the models, in general, treat diffusion – second order, fourth order, Rayleigh friction, etc.**

> A summary of approaches to both horizontal and vertical diffusion (in the free atmosphere) have been added (p.11, l.19 through p.12, l.9). We do not describe parametrizations of vertical diffusion in the boundary and surface layers, which differ more substantially, although we do include a reference to Chapter 2 of the S-RIP report where this information is provided.

**In both the text, Page 8, and the tables, the spatial size in km should be given for all of the models. Presently it is in degrees for some models and km for others. Consistent presentation of technical information should be checked throughout.**

> We chose this approach because grid sizes are invariant in degrees for regular grids but approximately invariant in km for reduced grids. We have considered including the grid spacing in km at the equator, but opted against it because this calculation is easy to do and we feel that including it in the table could dilute the intended emphasis on what it means for a system to use a regular grid versus a reduced grid.

**Coming from a numerical background, the equating of "resolution" with "grid cell size" always bothers me. Generally, I live with my idiosyncrasy. However, the statement in Section 3, Page 8, that the "effective horizontal resolution is . . ." is one I cannot let past. Since, effective resolution of weather and climate models is a topic of some controversy, with several lines of research, this is a term than should be used correctly. This is grid size, not the effective resolution, which will be 6 to 10 times the grid size.**

> Thank you for pointing this out. References to resolution here and in the column heading for Table 2 have been replaced with "grid spacing".

**I would like to see tabulated information on the treatment of CO2 and CH4 (Page 9) and aerosols (Page 10) in the radiative schemes. Since these reanalysis cover a span of time when climate is, definitively, not stationary, the treatment of the greenhouse gases seems fundamental. From the text, the treatment varies greatly from model to model, and I can't differentiate one treatment from another in the current text. As with the treatment of the upper layers of the model, I would expect the treatment of greenhouse gases and aerosols to have significant impact upon the science-based interpretation of the systems. Hence a quick summary table would be useful.**

> We have added a figure (Fig. 4) that summarizes the various treatments of $CO_2$, $CH_4$, and TSI, and revised the text to better clarify which aerosol (p.12, l.28–31) and greenhouse gas (p.13, l.14–23) boundary conditions correspond to which systems. We have also added a brief description of the aerosol assimilation used in MERRA-2 (p.13, l.1–7), which was omitted from the original submission.

**Similar to previous comment, I would like to see more description, table perhaps, of the sea-ice treatment in the models (Page 10). The changes in Arctic are large. There are trends in area of polar isotherms in the upper troposphere and lower stratosphere; more details of the treatment of the Arctic boundary conditions are needed.**

> We have added a table (Table 4) listing the sources of SST and sea ice boundary conditions used by the various reanalyses, along with grid sizes, temporal resolutions, and reference information where available. We have also added a reference to Bosilovich et al. (2015), who discussed these differences in more detail and prepared a summary figure that includes most (but not all) of the reanalysis systems examined in this paper.

**In the sentences, page 13, the use of the word "attempt" to describe the 3D-Var and 4D-Var assimilation systems is peculiar. I assume that they do, in fact, do the optimization calculation.**

> The word "attempt" has been removed.

**Page 15. Like sea-ice, I could imagine a tabulation of how water vapor (Page 21) is treated in the upper troposphere and stratosphere would be useful for science-based interpretation.**

> The current text hits all the major points (see also new paragraph from p.25, l.10–17). A tabular presentation of the same material will be available in Chapter 2 of the S-RIP report.

**Just by chance, I just saw that Jim Pfaendtner's name is misspelled in the Helfand reference (Page 31). It is spelled as "Pfaentner," and it should be "Pfaendtner."**

> This has been fixed, along with several other typos and errors.

**List of relevant changes**
(all page and line numbers are for the "track changes" version of the manuscript)

p.1, l.6: Andrea Molod (NASA GMAO) has been added as a co-author
p.2, l.27: removed parenthetical, replaced by the final two sentences of this paragraph (l.29–31), as suggested by the editor in charge
p.3, l.10–19: revisions to more explicitly acknowledge the SPARC Intercomparison of Middle Atmosphere Climatologies (requested by referee #2) and reduce the number of recent studies cited (requested by referee #1)
p.3, l.31–p.4, l.7: revisions to generalize the S-RIP programmatic material (response to referee #1)
p. 4, l.24: changed dates to account for MERRA-2 (which starts in 1980) and CFSR (which switches to CFSv2 at the end of 2010) – some data and diagnostics will not be available outside of this 1980–2010 range
p.7, l.12 & l.16: added and altered references for MERRA-2 for accuracy
p.8, l.14–15 & l.30: revisions to improve specificity (as suggested by referee #2)
p.9, l.12–p.10, l.2: added discussion of treatment of upper layers in the reanalysis forecast models (sponge layers, etc.), as suggested by referee #2
p.11, l.22–p.12, l.12: added representations of horizontal and vertical diffusion in the reanalysis forecast models, as suggested by referee #2
p.12, l.22–27: revisions to incorporate the new Table 4, which lists SST and sea ice lower boundary conditions used in the reanalyses (as suggested by referee #2)
p.12, l.29–p.13, l.1: edited to clarify which reanalysis prescribes which aerosol climatology, as suggested by referee #2.
p.13, l.5–l.11: added brief description of aerosol analysis in MERRA-2, which was omitted from the original manuscript
p.13, l.17–27: expanded description of key differences in radiatively active trace gas climatologies, as suggested by referee #2, and introduced upper two panels of the new Fig. 4, which illustrate differences in the $CO_2$ and $CH_4$ climatologies used in reanalyses
p.13, l.29: corrected error: MERRA uses a constant total solar irradiance (TSI), rather than the varying TSI produced by the SOLARIS working group for CMIP5
p.13, l.34–p.14, l.3: noted that the Total Irradiance Monitor correction is applied to TSI variations in both ERA-20C and MERRA-2, and corrected ranges accordingly
p.14, l.10–14: introduced the bottom panel of the new Fig. 4, which illustrates differences in TSI (omitting seasonal variations, which are larger in amplitude but less different amongst reanalyses)
p.16, l.10–11: removed "attempts to" (as suggested by referee #2)
p.16, l.30: removed reference to ERA-20C Ensemble, which is not discussed in this manuscript
p.18, l.20–21: added description of homogenization (as suggested by referee #1)
p.20, l.33–p.21, l.8: added discussion of some issues associated with the TOVS-to-ATOVS transition (partial response to referee #1)
p.22, l.32–p.23, l.1: brief explanation of why GNSS-RO data are assumed to be effectively bias-free (as suggested by referee #1)
p.25, l.15–22: added discussion of how the simple parametrizations affecting water vapour in the upper troposphere and stratosphere may influence the performance of the reanalysis system (as suggested by referee #1)
p.25, l.25–27: added references to relevant sections of the S-RIP interim report
p.26, l.6–p.27, l.15: summary and additional discussion of how aspects of reanalysis systems can introduce or exacerbate biases in reanalysis products (as suggested by referee #1)
p.48: removed table listing chapters of the S-RIP report (as suggested by referee #1)
p.50: replaced "resolution" with "grid spacing" (as suggested by referee #2)
p.52: added list of SST and sea ice data sets used as boundary conditions by the various reanalysis systems, along with reference information where available (as suggested by referee #2)
p.56: added figure showing time series of prescribed concentrations of $CO_2$ and $CH_4$ (under the globally well-mixed assumption) and prescribed TSI (omitting seasonal variations), as suggested by referee #2

[revised manuscript text omitted]

---

## Author Response (AR2)

Dear Dr. Peter Haynes:

Thank you very much for handling our manuscript and for your suggestions. Please see below our response.

*Both referees reports for the original version of this paper were positive one was 'accept after technical corrections' and the other was 'accept after minor revision'. The revision has responded thoroughly to the referees' comments. and the revised paper is improved significantly as a result. I don't see any need to return the revised paper to the referees for further comment.*

Thank you very much for your evaluation.

*The only significant point raised by a referee that has not been acted on is that of whether 'programmatic' material should be included in an ACP paper. Whilst inclusion of this type of material might not fit a traditional standard scientific paper, I don't think that one should be too dogmatic about this. The important question is whether inclusion of this material is potentially valuable to the readership over some decent time scale (e.g. it won't seem out-of-date in a year's time). So generally I think it is fine to retain this material.*

*However thinking about this when looking through the revised paper made me think of the following points: (i) the SRIP report is referred to several times as if it exists already, e.g. in Section 1 you have 'The S-RIP report consists of two parts.' (which replaces text including 'the planned SRIP report'). (ii) There is a bit of a confusion between 'project' and 'report' -- e.g. 'The SRIP project focuses predominantly on reanalyses, although some chapters include diagnostics from operational analyses when appropriate.' -- perhaps that should be 'some chapters of the planned report'. and (iii) the paper is described as 'an overview paper for the SRIP special issue' -- but what that means is not that it is an 'overview of the special issue' but that it is an 'overview of the SRIP project to be included in the special issue'.*

Text in the abstract, introduction, and summary sections has been revised following your suggestions. We also found three places in the reanalysis system description sections that needed revisions, along with the caption of Figure 1.
(i) "planned" has been added to "report(s)" in most cases, and we explicitly refer to "two (i.e., interim and full) reports" rather than two stages of the same report.
(ii) for "chapters", "of the planned reports" has been added in most cases.
(iii) the term "overview" is now used only as "overview of the reanalysis data sets" (or "systems"),

and is not used for the references to S-RIP programmatic material (project, report, special issue).

*All the above are fairly simple (perhaps trivial) points, but getting them (and more broadly the distinction between 'project', 'report' and 'special issue') absolutely transparent and clear would avoid reader confusion. It might be helpful to include somewhere a suitable paraphrase of the special issue details -- 'This special issue serves to collect research with relevance to SRIP in preparation for the publication of the SRIP report in 2018.'.*

This sentence, with some minor changes, has been added in the abstract and in the final paragraph of the introduction.

*A final point is that given the reference to the future SRIP report (which I think is useful) and your statement that both interim and final reports will be available in the SPARC report series, it might be helpful to say explicitly that these reports are available on the SPARC website and to give at least the address of the SPARC website as a whole.*

This has been written explicitly in the first paragraph of the summary and outlook section.

*Subject to consideration of the above I am pleased to accept the paper for publication in ACP. I've specified this as a further minor revision in case any dialogue between us is helpful but I expect to accept as is the next version of the paper.*

Thank you again for your very useful suggestions. Please feel free to let us know if you find other needed revisions and/or wish to propose better phrasing.

Please see below for a track-change version where you can see all the changes we have made.

Masatomo Fujiwara and Jonathon Wright on behalf of the author team for this paper.

[revised manuscript text omitted]